# Land management shapes drought responses of dominant soil microbial taxa across grasslands

J. M. Lavallee ●[1,2] ✉, M. Chomel ●[1,3], N. Alvarez Segura ●[4,5], F. de Castro ●[6,7], T. Goodall ●[8], M. Magilton ●[6,9], J. M. Rhymes ●[1,10], M. Delgado-Baquerizo ●[11,12], R. I. Griffiths[8,13], E. M. Baggs ●[14], T. Caruso ●[15], F. T. de Vries ●[1,16], M. Emmerson[6], D. Johnson ●[1] & R. D. Bardgett ●[1]

Soil microbial communities are dominated by a relatively small number of taxa that may play outsized roles in ecosystem functioning, yet little is known about their capacities to resist and recover from climate extremes such as drought, or how environmental context mediates those responses. Here, we imposed an in situ experimental drought across 30 diverse UK grassland sites with contrasting management intensities and found that: (1) the majority of dominant bacterial (85%) and fungal (89%) taxa exhibit resistant or opportunistic drought strategies, possibly contributing to their ubiquity and dominance across sites; and (2) intensive grassland management decreases the proportion of drought-sensitive and non-resilient dominant bacteria—likely via alleviation of nutrient limitation and pH-related stress under fertilisation and liming—but has the opposite impact on dominant fungi. Our results suggest a potential mechanism by which intensive management promotes bacteria over fungi under drought with implications for soil functioning.

Soil microbial communities mediate ecosystem functions including nutrient cycling, organic matter decomposition, and pathogen control[1–3], but their functioning can be impacted by climate extremes[4,5] which are becoming increasingly common. Recent evidence shows that despite very high diversity of soil microbial taxa, a small proportion can be considered dominant, i.e., they are found across most soils and are highly abundant relative to other taxa[6,7]. These dominant taxa may be drivers of ecosystem responses to climate extremes (i.e., the mass-ratio hypothesis[8]), an idea supported by studies of plant communities linking ecosystem responses to the abundances of dominant plant species[9,10]. Therefore, understanding how dominant microbial taxa respond to climate extremes and how these responses are shaped

[1]Department of Earth and Environmental Sciences, The University of Manchester, Oxford Road, Manchester M13 9PT, UK. [2]Environmental Defense Fund, 257 Park Ave S, New York, NY 10010, USA. [3]FiBL France, Research Institute of Organic Agriculture, 26400 Eurre, France. [4]Institute of Biological and Environmental Sciences, University of Aberdeen, St Machar Dr, Old Aberdeen, Aberdeen AB24 3UL, UK. [5]EURECAT—Centre Tecnològic de Catalunya, C/ de Bilbao, 72, 08005 Barcelona, Spain. [6]School of Biological Sciences and Institute for Global Food Security, Queen's University of Belfast, 19 Chlorine Gardens, Belfast BT9 5DL, UK. [7]AgriFood & Biosciences Institute, 18a Newforge Ln, Belfast BT9 5PX, UK. [8]UK Centre for Ecology & Hydrology Wallingford, Maclean Building, Benson Lane, Crowmarsh Gifford, Wallingford, Oxfordshire OX10 8BB, UK. [9]School of Life Sciences, University of Lincoln, Brayford Pool, Lincoln LN6 7TS, UK. [10]Centre for Ecology & Hydrology Bangor, Environment Centre Wales, Deiniol Road, Bangor LL57 2UW, UK. [11]Laboratorio de Biodiversidad y Funcionamiento Ecosistémico. Instituto de Recursos Naturales y Agrobiología de Sevilla (IRNAS), CSIC, Av. Reina Mercedes 10, E-41012 Sevilla, Spain. [12]Unidad Asociada CSIC-UPO (BioFun). Universidad Pablo de Olavide, 41013 Sevilla, Spain. [13]School of Natural Sciences, Bangor University, Deiniol Rd, Bangor LL57 2UR, UK. [14]Global Academy of Agriculture and Food Systems, Royal (Dick) School of Veterinary Studies, Easter Bush Campus, Charnock Bradley Building, University of Edinburgh, Edinburgh EH25 9RG, UK. [15]School of Biology and Environmental Science, University College Dublin, Dublin, Ireland. [16]Institute for Biodiversity and Ecosystem Dynamics, University of Amsterdam, 1090 GE Amsterdam, Netherlands. ✉e-mail: jlavallee@edf.org

by environmental factors and land management will enable better predictions of ecosystem behaviour into the future[11,12].

Soil microbial taxa can be categorised by life history strategies[13,14] to inform on their capacity to resist and recover from climate extremes such as drought[11,15]. These life history strategies are thought to emerge from correlated sets of traits (e.g., related to resource acquisition, growth yield, and stress tolerance), which are favoured under different environmental conditions[14]. For example, soil microbial communities subjected to moisture pulses may have greater proportions of taxa exhibiting a stress-resistant strategy, whereas those under ambient conditions may have higher abundances of drought-sensitive taxa[13]. Land management may also shift microbial life history strategies by changing resource availability and plant communities—environmental factors known to shape microbial community structure and function[16–21]. However, the interacting effects of land management and climate extremes such as drought have not been studied in the context of microbial life history strategies. This is a necessary step towards using ecological knowledge of soil microbes to predict and understand the consequences of land management decisions on soil functioning and sustainability in the face of climate change.

Here, we carried out a large-scale field experiment across a broad range of grassland sites to explore how the relative abundances of dominant microbial taxa with different drought-response strategies are shaped by soil conditions, climate, and land management intensity. We imposed a simulated drought on 15 pairs of grasslands under contrasting management (i.e., intensive and extensive) in three geographically distinct regions of the UK representing a range of soil and climatic conditions (Fig. S1, Table S1). Using an operational approach, we identified dominant microbial taxa and classified them into three broad drought-response strategies (i.e., resistant [no detectable response], opportunistic [positive response], or sensitive [negative response])[13]. We examined the interacting effects of climate, soil properties, and historical grassland management on dominant microbial taxa by drought-response strategy immediately following the drought and after a 60-day post-drought period[22], to capture both microbial resistance (lack of response to a perturbation) and resilience (recovery to an un-perturbed state) to drought[23,24].

We hypothesised that: (1) dominant soil microbial taxa largely display resistant or opportunistic strategies under drought, because a capacity to withstand variable moisture conditions would partly explain their ubiquity and abundance across sites; (2) intensive grassland management, characterized by regular fertiliser and lime application and higher plant productivity (Table S1), favours taxa that are maladapted to low resource availability and stress and therefore sensitive to drought; and (3) intensive grassland management favours microbial taxa that recover after drought (i.e., resilient), because more favourable soil conditions allow drought-affected taxa to rebound quickly with rewetting.

Our results show that most dominant soil microbial taxa were resistant to drought, as expected. We further show that intensive grassland management increases the proportion of dominant bacterial taxa that are resistant or opportunistic in the face of drought relative to those that are sensitive, and increases the proportion of dominant bacterial taxa that are resilient relative to those that are not resilient. However, intensive management has the opposite effect on dominant fungal taxa, increasing the proportions of sensitive and non-resilient taxa. Our finding that land management shapes the drought-response strategies of dominant soil microbial taxa has important implications for microbial community structure and function. Intensive grassland management is known to broadly favour bacteria over fungi, impacting key functions including soil carbon and nitrogen cycling[25,26]; our results suggest this pattern may be exacerbated as droughts become more frequent and intense with climate change.

## Results

### Most dominant soil microbial taxa are resistant to drought

We found that a relatively small number of bacteria and fungi dominate soils across the grassland sites, and that these taxa were highly resistant to an imposed drought event. For bacteria, dominant taxa (defined as present across all 15 sites and in the top 10% of relative abundance ranked by 16S rRNA reads[7]) represented 1269 out of 19224 total operational taxonomic units (OTUs), which accounted for ~7% of total OTUs but 76% of all reads. For fungi, dominant taxa (present across all three regions and in the top 10% by ITS rRNA reads) made up 209 out of 12837 total OTUs, accounting for ~2% of total OTUs but 53% of all reads. Overall, the majority of dominant bacterial (66%) and fungal (64%) taxa were classified as displaying a resistant drought strategy, as they showed no response to drought in our hierarchical model using all data across sites and management regimes immediately after the simulated drought (Table S2). Opportunistic taxa, whose relative abundances increased in response to drought, represented 19% of dominant bacteria and 25% of dominant fungi; sensitive taxa, whose relative abundances decreased with drought, represented 12% of dominant bacteria and 7% of dominant fungi.

Dominant bacterial phyla in our dataset comprised primarily (by reads) *Proteobacteria* (32%), *Acidobacteria* (21%), *Verrucomicrobia* (13%), *Bacteroidetes* (11%), *Firmicutes* (9%), *Actinobacteria* (7%), *Chloroflexi* (3%), and several other globally distributed taxa. Of these phyla, most contained taxa representing each of the three drought-response strategies (Fig. 1). However, members of *Firmicutes* and *Bacteroidetes* tended to display resistant or sensitive drought-response strategies, with few or no taxa identified as opportunistic (zero out of 47 in *Firmicutes*; five out of 175 in *Bacteroidetes*). Members of *Acidobacteria, Actinobacteria*, and *Chloroflexi* tended to display resistant or opportunistic drought-response strategies, with few taxa identified as sensitive (nine out of 227 in *Acidobacteria*, one out of 118 in *Actinobacteria*, and one out of 65 in *Chloroflexi*). Dominant fungal phyla comprised (by reads) *Mortierellomycota* (48%), *Ascomycota* (22%), *Basidomycota* (15%), *Glomeromycota* (1%), and several other known and globally distributed or unidentifiable taxa. Members of *Ascomycota* tended to display resistant or opportunistic drought-responses strategies, with only six of 94 taxa identified as having a drought-sensitive strategy. Members of *Mortierellomycota, Basidomycota*, and *Glomeromycota* tended to display resistant or sensitive drought-response strategies, with only one or no taxa identified as opportunistic in each phylum (Fig. 1). Overall, dominant taxa resistant to drought belonged to different taxonomic groups dispersed across every major lineage of the phylogeny, suggesting that this capability is not limited to specific phylogenetic groups of microbes.

### Management affects dominant bacteria and fungi differently

We used structural equation models to infer potential mechanisms through which grassland management affected opportunistic, sensitive, and resistant dominant microbial taxa across sites (Fig. 2). Except for sensitive bacterial taxa, intensive management increased the relative abundances of all dominant microbial drought-response groups. Opportunistic and resistant bacterial taxa were positively impacted by intensive management at both timepoints (both directly and via increased pH; Fig. 2a, b), while sensitive bacterial taxa were either unaffected (following drought) or negatively affected (after the recovery period). Opportunistic and resistant fungal taxa were also positively affected by intensive management (either directly or via increased pH; Fig. 2c, d), but in contrast to sensitive bacterial taxa, sensitive fungal taxa were positively and directly affected by intensive management at both timepoints. As a result, the ratio of opportunistic:sensitive dominant taxa increased under intensive management for bacteria but decreased for fungi (Fig. 3a).

Of the environmental variables we considered in the SEMs (total C and N, temperature, texture, moisture, and pH), pH played the most

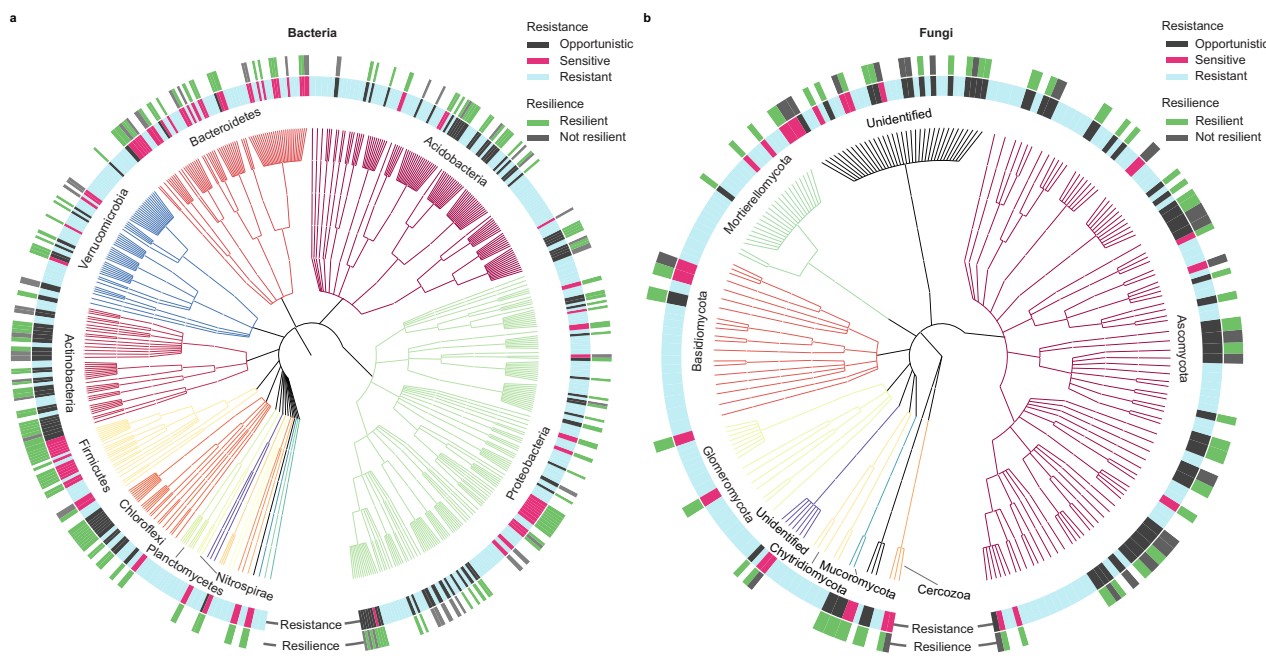

**Fig. 1 | Taxonomic tree showing drought responses of dominant soil microbial taxa.** Dominant bacterial (**a**) and fungal (**b**) community responses to drought immediately following the drought treatment ("resistance") and after the 60-day post-drought period ("resilience"), limited to the top 500 most abundant taxa across all samples for readability. The inner ring shows the taxonomic tree, coloured by phylum. The middle ring displays responses of each out immediately following drought (light blue = resistant, black = opportunistic, pink = sensitive). The outer ring displays responses after the 60-day post drought period (green = resilient taxa that recovered to control levels, dark grey = not resilient). Taxa defined as resistant to drought (light blue, inner ring) were not tested for resilience. Source data and identities of all OTUs are provided on GitHub[73].

important role. There were strong positive indirect effects of management intensity via increased soil pH for opportunistic and resistant bacterial taxa at both timepoints (Fig. 2a, b). Further investigation revealed unimodal relationships between pH and resistant and resilient bacterial taxa that peaked ca. pH 5.7 (Fig. S4). Fungal taxa were less impacted by pH overall, but there was a positive effect on opportunistic fungal taxa after the drought (Fig. 2c), and a negative effect on resistant fungal taxa after the recovery period (Fig. 2d). While the inclusion of pH did account for one mechanism by which management impacts microbial taxa, the fact that direct paths from the management variable manifested in the SEMs indicates that other mechanisms related to management (and not captured by total soil C and N, soil temperature, texture, and soil water content) are also impacting dominant microbial taxa in these soils. Intensive management did impact other key variables including above-ground plant biomass and plant-available N (Fig. 3) that are implicitly represented by our management variable in the SEM. In general, dominant fungal groups were impacted more strongly by the management variable in our SEMs, while dominant bacterial groups were impacted more strongly by pH and other soil characteristics (total soil C and N, soil temperature, texture, and soil water content).

### Drought treatment and soil moisture effects on dominant microbes

Drought treatment was the best predictor of soil moisture immediately after the simulated drought (day 0), with latitude and soil properties captured in the composite soil variable (total C and N, temperature, texture) also playing important roles (Fig. 2a, c). The drought treatment effect on the different microbial drought-response strategy groups was not fully captured by the field measurements of soil moisture—which only provided a snapshot of soil moisture conditions at the time of sampling—indicated by the direct paths from drought treatment for several microbial groups at that timepoint at day 0

(Fig. 2a, c). After the 60-day post-drought period, the drought treatment no longer predicted soil moisture or microbial drought-response strategy groups. Instead, latitude was a very strong predictor of soil moisture, and soil properties (composite soil variable) were an important predictor for bacterial drought response groups, but not fungal drought response groups (Fig. 2b, d). The absence of drought treatment effects on sensitive and opportunistic bacterial or fungal taxa after the 60-day post-drought period indicates group-level recovery within that time (Fig. 2b, d).

### Most drought-affected dominant bacteria and fungi are resilient

We categorized individual opportunistic and sensitive dominant taxa as resilient or not based on their abundances relative to ambient control plots after the 60-day post-drought recovery period. While most of the 503 drought-affected (opportunistic or sensitive) taxa were found to be resilient after 60 days, we identified 110 taxa that were not (Fig. 1). Of these, 34 were sensitive bacterial taxa and 8 were sensitive fungal taxa that differed from ambient control plot levels after the 60-day post-drought recovery period. Analyses of resilient bacterial and fungal taxa groups in control plots across both timepoints revealed that the relative proportion of resilient taxa (ratio of resilient:not resilient taxa; Fig. 3b) was higher for bacteria but lower for fungi under intensive compared to extensive grassland management.

## Discussion

Our study provides novel evidence, from a broad range of grassland sites varying in climatic and soil conditions (Table S1), that dominant soil microbial taxa are highly resistant to drought. Despite significant and sizable reductions in soil moisture under experimental drought across sites (Fig. S2), the majority of dominant soil microbial taxa either did not respond or responded positively. Of the taxa that were negatively impacted by the drought treatment (drought-sensitive strategies), the majority were resilient (i.e., did not differ from ambient

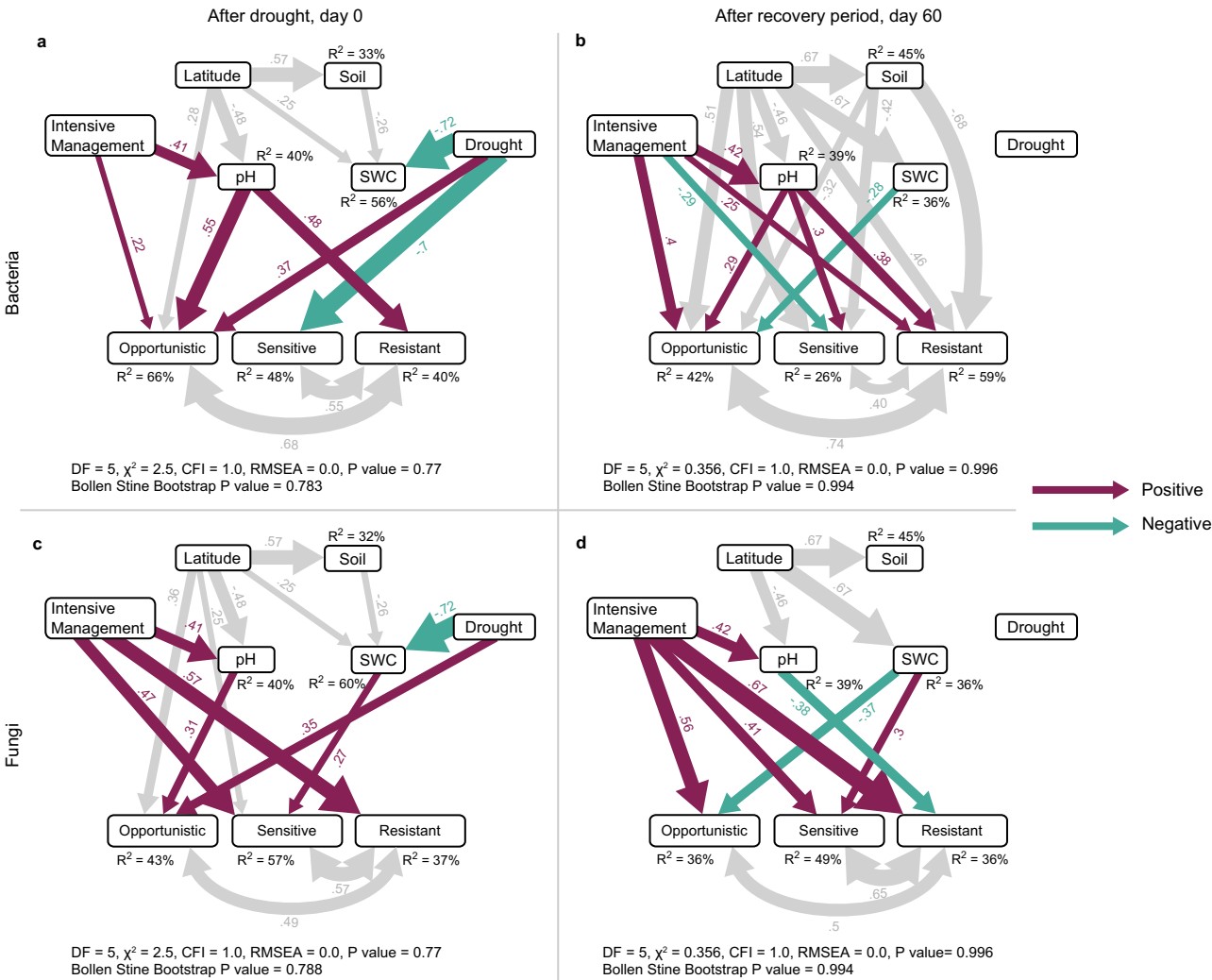

**Fig. 2 | Potential mechanisms potential mechanisms affecting dominant soil microbial taxa across sites.** Structural equation models (SEMs) of dominant bacteria (panels **a** and **b**) and fungi (panels **c** and **d**) at the two time points in this study: immediately following the drought treatment (day 0, panels **a** and **c**), and after the 60-day post-drought recovery period (panels **b** and **d**). The "soil" variable is a composite representation of soil C, N, texture, and temperature. SWC is soil water content by volume. The drought-response strategy (opportunistic, sensitive, resistant) of each OTU was determined by the drought treatment effect using linear mixed models. Arrow (path) thickness corresponds to the standardized coefficients, also written next to their respective paths. Paths of less interest are shaded grey to improve overall readability. Wald tests were used to evaluate the null hypothesis that individual path coefficients were equal to zero. Solid arrows indicate individual path coefficients with $P$-values < 0.05; path coefficients with $P$-values > 0.05 are not shown. Overall model fit for each model was assessed using Chi-squared test ($\chi^2$) with associated $P$-value and degrees of freedom (DF), Comparative Fit Index (CFI), and root mean square error of approximation (RMSEA). See Fig. S5 and Supplemental Note 1 for more detail. Source data are provided on GitHub[73].

control levels within the 60-day post-drought period). The resistance and resilience of these soil microbial taxa to drought, observed here across three geographically distinct regions of the UK, may in part explain why they are present and highly abundant (i.e., dominant) across sites[7]. The use of a distributed landscape design combined with an in situ experimental drought treatment uniquely demonstrates that responses of dominant soil microbial taxa to drought are consistent at a large spatial scale. Though drought severity can be difficult to quantify[27], especially at the microscale most relevant to microbiota[28], we observed significant effects of the drought treatment on ecosystem respiration and microbial community structure (including non-dominant taxa) at the plot scale across all regions, indicating that our drought treatment was ecologically significant (Fig. S2, Fig. S3, Table S3). Our findings align with recent studies showing that abundant microbial taxa are more resistant to perturbations[29], are adapted to broader ranges of environmental conditions[30,31], and display higher frequencies of genomic traits associated with stress-tolerance and competitive abilities[6] than rare microbial taxa. These results suggest that dominant microbial taxa in grassland soils are generalists adapted to varying environmental conditions, allowing them to withstand perturbations and thrive across a broad range of sites.

We found that environmental context and land management did affect the relative abundances of dominant microbial taxa with different drought response strategies, but not in ways we expected. We expected that the impacts of grassland management on the resistance and resilience (i.e., the capacity to recover) of dominant microbial taxa to drought would be inversely related[32,33], and that bacterial and fungal communities would respond similarly. More specifically, we hypothesised that microbial communities in intensively managed grasslands would be more sensitive to drought due to lower stress-tolerance but be more resilient due to higher available nutrients and more ideal pH levels enabling recovery. However, our findings suggest that resistance and resilience of dominant soil microbial taxa are positively related in the context of grassland management, and that bacterial and fungal

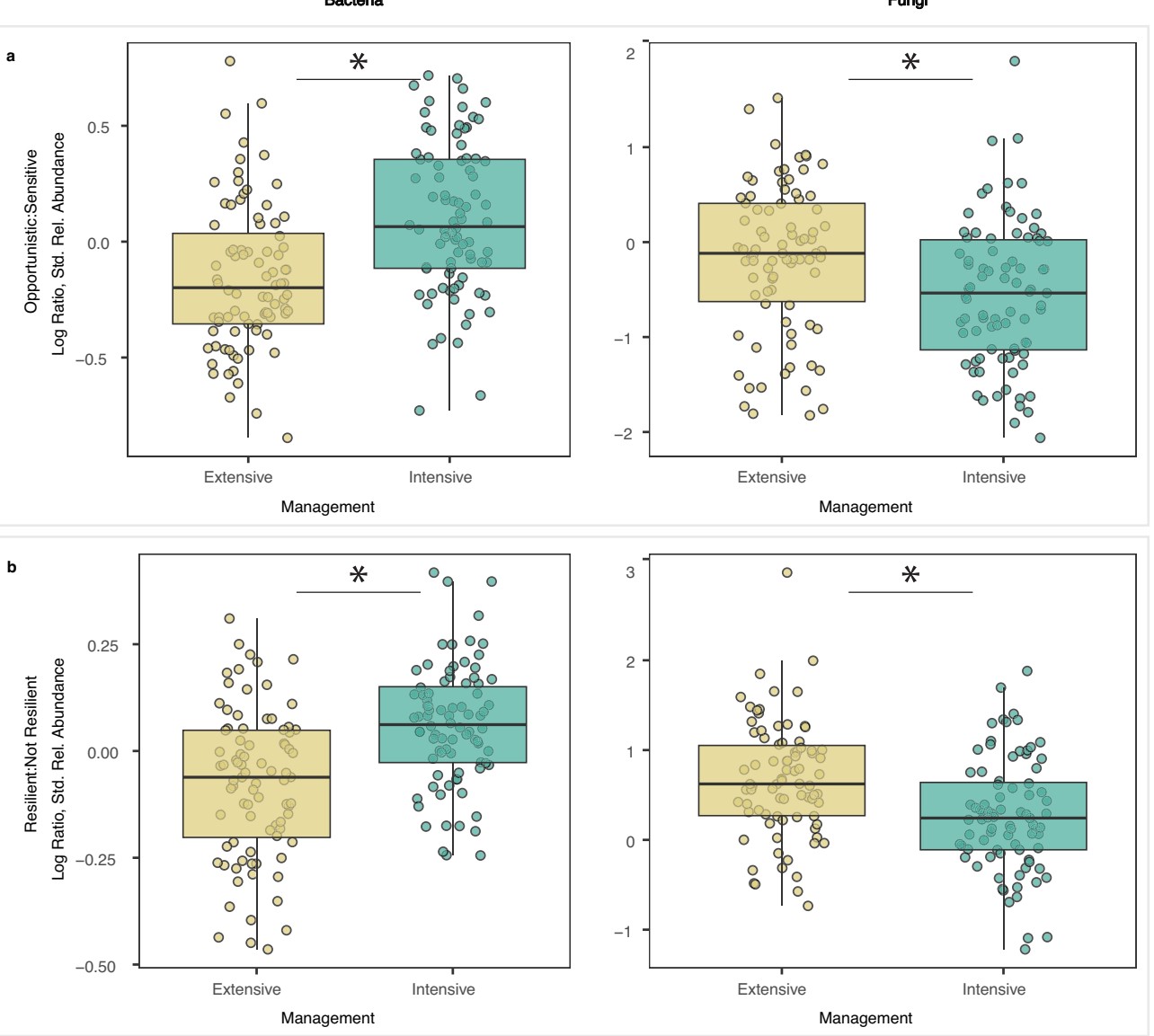

**Fig. 3 | Grassland management effects on dominant soil microbial taxa.** Ratios of the standardized relative abundances of opportunistic:sensitive dominant taxa (**a**) and resilient:not resilient dominant taxa (**b**) by grassland management, with bacteria on left and fungi on right. Boxplots show the median (centre line), first and third quartiles (box limits), and smallest and largest values within 1.5x interquartile range (whiskers), and all datapoints are shown ($n$ = 90 experimental plots for all boxplots). Resilient taxa were defined as having similar relative abundances to those in control plots after the 60-day post-drought recovery period. Ratios were higher for bacteria but lower for fungi in intensively managed grasslands based on linear mixed models using data from control plots across both timepoints (*P*-value < 0.05 indicated by *). Output of linear mixed models of effects of intensive versus extensive management for each response variable: (**a**), left: t(14) = 5.61, *P* = 0.0001, effect size = 0.277, 95% Confidence Intervals = 0.267, 0.287; panel a, right: t(14) = −4.58, *P* = 0.0004, effect size = −0.382, 95% Confidence Intervals = −0.542, −0.222; (**b**), left: t(14) = 3.48, *P* = 0.0036, effect size = 0.134, 95% Confidence Intervals = 0.058, 0.21; panel b, right: t(14) = −2.20, *P* = 0.045, effect size = −0.377, 95% Confidence Intervals = −0.717, −0.037. Source data are provided on GitHub[73].

communities respond to intensive and extensive grassland management in divergent ways. Compared to communities under extensive management, dominant bacterial communities under intensive management shifted toward less sensitive and more resilient drought strategies, while dominant fungal communities shifted toward more sensitive and less resilient drought strategies (Fig. 3). This suggests that across these grassland sites, dominant bacterial communities under more intensive management are better able to withstand and recover from drought than those under extensive management, while dominant fungal communities are not. Again, these findings were apparent when data were aggregated across all three UK regions, which cover a broad range of climatic and soil conditions.

The divergence between bacterial and fungal responses to more intensive management may be explained by differences in their sensitivities to prevailing conditions including pH, nutrients, and plant productivity. The intensively managed grasslands used in our study all receive regular inputs of inorganic fertilisers to reduce nutrient limitation along with lime, which increases pH toward neutral levels and leads to increased plant productivity (Fig. 4). For the dominant bacterial communities at these sites, liming likely alleviates pH-related stress, allowing opportunistic taxa to succeed under the drought treatment relative to other taxa. These opportunistic taxa may have traits related to high growth yields or efficient resource acquisition[34] that enable them to take rapid advantage of abundant resources under

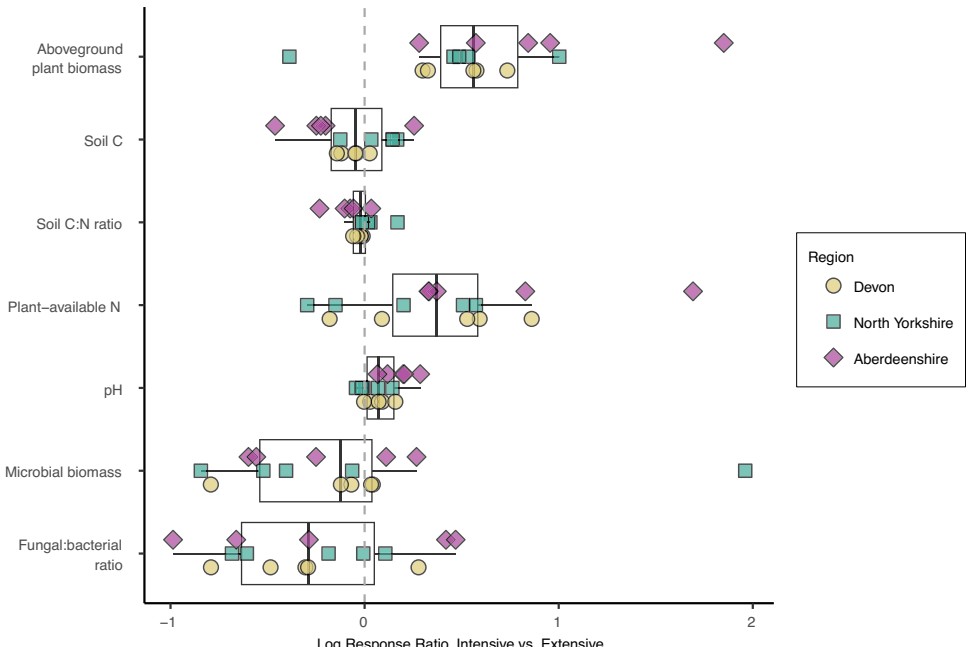

**Fig. 4 | Grassland management affects a range of environmental variables.** Log response ratio of key variables related to grassland intensification (C is carbon, N is nitrogen). Log response ratio is calculated as the natural log of the ratio of the value of a given variable in an intensively managed field to the corresponding value in the paired extensively managed field. Fifteen pairs of intensive and extensive grasslands, 5 per region, are shown here with each point representing the log ratio of within-field means for one pair (site). Boxplots show the median (centre line), first and third quartiles (box limits), and smallest and largest values within 1.5x interquartile range (whiskers), and all datapoints are shown. Source data are provided on GitHub[73].

changing conditions. Indeed, the higher soil pH observed in the intensively managed grasslands (due to lime application) positively affected resistant and resilient bacterial taxa relative to the extensive grasslands with more acidic soils (Fig. S4). This finding agrees with previous work on similar soils suggesting that relief from acidic conditions allows bacterial communities to shift from maintenance to growth strategies[35]. In that study, the key pH threshold for shifts in microbial strategies was found to be pH ca. 6.2, however, in our study the pH in intensively managed fields rarely surpassed that threshold, suggesting the pH threshold could be lower for many of our sites. In addition to higher pH, the higher soil nutrient availability and plant productivity in the intensively managed grasslands likely further favoured copiotrophic or high-yield bacterial taxa[36] capable of taking advantage of changing conditions under drought, or capitalizing on flushes of nutrients upon rewetting of the droughted plots[20,34]. Indeed, *Actinobacteria* had the highest proportion of opportunistic taxa in our study (consistent with a previous large-scale study of drought effects on microbial communities in grasslands[4]) and this phylum is thought to comprise primarily copiotrophs or high-yield strategy taxa favoured by N additions[20,36–38]. Further, *Verrucomicrobia* and *Acidobacteria*, which that are thought to be comprised of mainly oligotrophs (taxa that grow slowly and perform well under nutrient-poor conditions relative to copiotrophs)[20,37,38], had the lowest proportions of drought-sensitive taxa that were resilient.

In contrast to dominant bacterial communities, dominant fungal communities under more intensive management generally displayed lower resistance and resilience to drought than in extensively managed grasslands. Fungal communities are known to be less sensitive to pH than bacteria[39], and we didn't observe strong pH effects on resistant or resilient dominant fungal taxa in this study (Fig. S4), suggesting that alleviation of pH-related stress was not as relevant a mechanism for fungi in this case. Instead, other local-scale impacts of management such as increased plant biomass and available nutrients (Fig. 4) were the likely drivers of fungal responses, as suggested by the fact that the management variable in our SEMs generally affected dominant fungal groups more strongly than pH or prevailing soil conditions (Fig. 2). Fungal communities have been shown to respond strongly to fertilisation[20,40] and are often suppressed relative to bacteria under more intensive grassland management[17,41,42], consistent with our observation of lower fungal:bacterial ratios under intensive compared to extensive management across sites (Fig. 4). Furthermore, we recently showed in a sub-set of the grassland sites studied here that intensive management reduces the flux of recent photosynthate to soil food webs including arbuscular mycorrhizal fungi, indicating importance of this pathway for driving fungal activity[43]. It is therefore possible that this pathway of reduced energy flux could contribute to the increased sensitivity of dominant fungal communities to drought (which further reduces the flux of recent photosynthate belowground[44]) in intensively managed grasslands. The opposing responses of dominant bacteria and fungi to grassland management in terms of their resistance and resilience to drought may help to explain widespread observations of decreasing fungal:bacterial biomass ratios with grassland intensification[17,42,45].

Overall, the alignment of resistance and resilience in the context of grassland management intensity for both bacteria and fungi was unexpected, as other studies have found trade-offs between resistance and resilience in soil microbial communities[33,46,47]. However, in our study encompassing a relatively broad range of soils, pH was an important driver of both resistance and resilience in dominant bacterial communities, while fertilisation may have driven both resistance and resilience of dominant fungal communities, which would help to explain the alignment in both responses with management. While consistencies in taxon-level responses to separate drought and nitrogen addition treatments has been observed previously[48], the sets of traits determining responses to soil water availability versus nutrient availability or pH may not always align and a multi-dimensional framework may be necessary for considering microbial life history strategies[49] and predicting microbial responses to climate extremes.

We observed contrasting phylum-level responses to drought in soil dominant bacterial and fungal communities, suggesting that certain phyla may be inherently more resistant and resilient to drought than others. *Actinobacteria* contained a high proportion of resistant and opportunistic taxa, with only one taxon identified as sensitive, consistent with previous observations that *Actinobacteria* are prevalent in dry environments[50] and are highly resistant or increase in response to drought[4,49]. Members of *Firmicutes* and *Bacteroidetes* were generally more sensitive to the drought treatment, and while members of *Bacteroidetes* have been shown to decrease in relative abundance in drier soils, members of *Firmicutes* have previously shown the opposite response[51]. It is possible that these previous observations may have been driven primarily by one or a few taxa, which may not have been present (or defined as dominant) here. Context-dependent drought responses have been previously observed for other phyla including *Proteobacteria* and *Planctomycetes*[51], and we also observed relatively high numbers of taxa with different drought-response strategies in those phyla.

Dominant members of *Ascomycota* were particularly opportunistic under drought, which agrees with findings that *Ascomycota* are dominant globally and are generalists that are adapted to a wide range of conditions[6,50]. Within *Glomeromycota*, dominant taxa that responded to our drought treatment were sensitive, in agreement previous findings that both community composition[4] and functionality[52,53] of this group of fungi respond to drought in other systems. However, the majority of dominant *Glomeromycota* in this study were found to be resistant to the drought, suggesting again that the results from these other studies my largely be driven by only a few dominant members of *Glomeromycota*, or by taxa that were not defined as dominant here. Two members of *Basidiomycota* indicated sensitivity to drought, and one taxon did not recover after the 60-day post-drought period. Given that *Basidiomycota* are important decomposers and ectomycorrhizal symbionts in forests[54], microbial communities in forested systems (or under forest expansion) may be sensitive to drought with potential implications for forest growth and ecosystem functioning[55], which deserves further study.

Overall, our findings from a broad range of grassland sites across the UK indicate that most of the dominant soil microbial taxa are highly resistant to drought, which may explain their prevalence across a diverse range of grassland soils. We further show that grassland management, along with climate and soil properties, shapes the relative abundances of dominant soil microbial taxa with differing drought-response strategies. More intensive grassland management, which creates more optimal pH and higher nitrogen availability compared to extensive management, promotes opportunistic and resilient bacterial taxa that may employ copiotrophic or fast-response strategies and are able to take advantage of changing conditions. However, it has the opposite effect on dominant fungal taxa which may help to explain increases in bacterial prevalence over fungi with grassland intensification[17,42,45,56]. Our results suggest the pattern of bacterial prevalence over fungi under intensive management may be reinforced or exacerbated as droughts become more frequent and intense with climate change, and potentially contribute to less efficient carbon and nitrogen cycling in these systems[25,26].

By demonstrating that land management shapes the drought-response strategies of dominant microbial taxa across grasslands, our findings improve our understanding of how soil microbial communities respond to drought. Moreover, by identifying consistent management- and drought-induced responses of dominant microbial taxa, our findings pave the way for future studies that interrogate their functional attributes and links to key ecosystem functions[57]. Given the enormous complexity of soil microbial communities and their dynamics in space and time, our approach of focusing on the drought response strategies of dominant taxa is one way to make this task more feasible in the future.

## Methods

### Field sites

The field experiment was carried out between May and September of 2016 across a series of mesotrophic grasslands in the United Kingdom, concentrated in three regions: Devon in southwest England, North Yorkshire in northern England, and Aberdeenshire in northeast Scotland (Fig. S1, Table S1). Prior to the start of the experiment, we identified 15 pairs of fields on working farms with contrasting management and classified them as either intensively or extensively managed based on observations of plant communities and interviews with farmers and land managers. Extensively managed fields received very low or no synthetic fertiliser and lime, had more diverse plant communities, were generally not cut for hay or silage, and were grazed at low stocking densities by sheep or cattle. Intensively managed fields received regular applications of fertiliser and lime (as deemed necessary by the farmer), had less diverse plant species mixtures, were cut for hay or silage, and were grazed at higher stocking densities. Differences in management had been maintained for at least 10 years, and typically longer (Table S1). Wherever possible, we identified paired intensive and extensive fields that were adjacent, to minimize differences in intrinsic environmental variables such as topography, weather patterns, and soil type. If fields were not immediately adjacent, we chose fields no more than 0.5 km apart and used farmer and land manager interviews to ensure minimal differences between paired fields aside from management.

### Experimental design

This study employed a randomized complete block design with subsampling. In each region, 5 sites were identified that each had two differently managed fields within 0.5 km for a total of 15 sites and 30 paired fields. In each of the 30 fields, three pairs of drought and ambient control plots were established and enclosed in fencing for protection from large mammals and machinery. A field drought was simulated by placing a transparent roof (1.5 m * 1.3 m) on each drought plot alongside its paired delimited control plot for 60 days between May and July of 2016, which equates to a >100-year drought for these sites[58]. In total, there were 90 pairs of droughted and ambient control plots, and the three within-field replicates of each were treated appropriately in all statistical models by either including site and field as random effects or by aggregating the data at the field scale where random effects could not be modelled. At the end of the drought period, drought shelters were removed and an initial ("day 0") sampling and measurement of soil functions (in both drought and control plots) was carried out to assess the impact of the drought relative to the ambient control conditions. Sampling and measurement were done in the centre of the plots, leaving a 15 cm buffer to minimize edge effects. Immediately following this sampling event, droughted plots were watered (amounts were based on average July rain events from 2007-2011 for the nearest Met Office from each region[59]) to stimulate the start of the post-drought period. Sampling of drought and ambient control plots was repeated 60 days after the removal of the shelters to capture recovery during the post-drought period (resilience).

### Soil sampling

At all timepoints, multiple soil samples were collected to 10 cm depth and composited for measurements of soil nematode communities (6 * 1.3-cm diameter cores) soil microarthropod communities (4 * 2.5-cm diameter cores), and soil microbiota and chemical analysis (3 * 2.5-cm diameter cores). Soil samples were immediately composited in plastic sample bags and transferred to coolers for transport to laboratories within 24–48 h. Samples intended for soil fauna analysis were kept open to allow for gas exchange. At each sampling event, soil moisture and temperature were measured using Wet Sensor probes (WET-2, Delta-T Devices, Cambridge, UK). Bulk density was measured using the core technique at the time of the drought treatment establishment, using

one core per plot for a total of 6 cores per field, and the average value for each field was used throughout the study.

## Soil biogeochemical analysis

Samples for analysis of microbial communities, texture, and C and N analyses were transported to the University of Manchester and stored at 4 °C for a maximum of 3 days until further processing and analysis. All samples were sieved to 4 mm for homogenization and removal of visible plant material and rocks, after which samples were divided for further analyses. One subsample was immediately frozen at −80 °C awaiting microbial DNA sequencing. A second subsample was weighed, placed in a paper bag, and dried to constant weight at 40 °C to calculate soil moisture. This subsample was used for further analyses of total C and N concentrations using a Vario Cube (Elementar Americas Inc., Ronkonkoma, NY, USA), and soil texture analysis by laser granulometry using a Malvern Mastersizer 2000 (Malvern Instruments Ltd, Malvern, Worcestershire, UK) following removal of organic matter with $H_2O_2$ at 50 °C overnight. Soil pH was measured on field moist subsamples in slurries of 1:2.5 soil:deionized water using a pH meter (Seven2GO Mettler Toledo, Columbus, Ohio, USA). Further analyses are described in Supplementary Methods.

## 16S and ITS amplicon sequencing and data analysis

Amplicon sequencing and bioinformatic and statistical analyses of sequencing data were done following the methods of De Vries et al.[60] DNA was extracted from 0.16 g of soil using the MoBIO PowerSoil-htp 96-Well DNA Isolation kit (Carlsbad, CA, USA) according to the manufacturer's protocols and the DNA quality was checked by agarose gel electrophoresis. Bacterial 16S rRNA sequencing followed the dual indexing protocol of Kozich et al.[61] for the MiSeq plaform (Illumina, San Diego, CA, USA). Each primer consisted of the appropriate Illumina adaptor, 8-nt index sequence, a 10-nt pad sequence, a 2-nt linker, and the amplicon specific primer. The V3–V4 hypervariable regions of the bacterial 16S rRNA gene were amplified using primers 341 F[62] and 806 R[63], CCTACGGGAGGCAGCAG, and GCTATTGGAGCTGGAATTAC, respectively. Amplicons were generated using high-fidelity DNA polymerase Q5 Taq (M0491L, New England Biolabs, Ipswich, USA), premixed dNTPs (BIO-39053, Meridian Bioscience, Ohio, US), and using Eppedorf Mastercycler Nexus PCR machines (Hamburg, Germany). After an initial denaturation at 95 °C for 2 min, PCR conditions were: denaturation at 95 °C for 15 s, annealing at 55 °C for 30 s with extension at 72 °C for 30 s, repeated for 30 cycles, followed by a final extension of 10 min at 72 °C.

Fungal internal transcribed spacer (ITS) amplicon sequences were generated using a 2-step amplification approach. Primers GTGARTC ATCGAATCTTTG and TCCTCCGCTTATTGATATGC[64] were each modified at the 5′ end with the addition of Illumina pre-adaptor and Nextera sequencing primer sequences. After an initial denaturation at 95 °C for 2 min, PCR conditions were: denaturation at 95 °C for 15 s, annealing at 52 °C for 30 s with extension at 72 °C for 30 s, repeated for 25 cycles, with a final extension of 10 min at 72 °C included. PCR products were cleaned using a DNA Clean-up Kit (ZR-96, Zymo Research Inc., Irvine, US) following manufacturer's instructions. MiSeq adaptors AATGAT ACGGCGACCACCGAGATCTACAC and 8nt dual-indexing barcode sequences were added during a second step of PCR amplification. After an initial denaturation 95 °C for 2 min, PCR conditions were: denaturation at 95 °C for 15 s; annealing at 55 °C for 30 s with extension at 72 °C for 30 s; repeated for 8 cycles with a final extension of 10 min at 72 °C.

Amplicon concentrations were normalized using SequalPrep Normalization Plate Kit (A10510-01, Thermo Fisher Scientific, Waltham, US) and amplicon sizes determined using an 2200 TapeStation (Agilent, Santa Clara, US) prior to sequencing each amplicon library separately using MiSeq (Illumina, San Diego, US) with V3 600 cycle

reagents (MS-102-3003, Illumina, San Diego, US) at concentrations of 14 and 7 pM (16S and ITS respectively) with a 5% PhiX control v3 (FC-110-3001, Illumina, San Diego, US) library.

Sequenced paired-end reads were joined using PEAR[65], quality filtered using FASTX tools (hannonlab.cshl.edu), and length-filtered to a minimum length of 300 bp. The presence of PhiX and adaptors were checked for and removed with BBTools (jgi.doe.gov/data-and- tools/ bbtools/), and chimeras were identified and removed with VSEARCH_UCHIME_REF[66] using Greengenes Release 13_5 (at 97%). Singletons were removed and the resulting sequences were clustered into operational taxonomic units (OTUs) with VSEARCH_CLUSTER[66] at 97% sequence identity. Representative sequences for each OTU were taxonomically assigned by RDP Classifier with the bootstrap threshold of 0.8 or greater using the Greengenes Release 13_5 (full) as the reference. Unless stated otherwise, default parameters were used for all steps listed. The fungal ITS sequences were analysed using PIPITS[67] with default parameters. Briefly, this involved quality filtering and 97% clustering of the ITS2 region as indicated above for the 16S processing, using the UNITE database for chimera removal and taxonomic identification of representative OTUs. Both bacterial and fungal OTU abundance tables were rarified to a minimum of 9000 reads per sample, and samples with zero reads were removed prior to further analyses.

Plots showing circular representations of the taxonomic trees were created using the GraPhlAn software tool (https://huttenhower. sph.harvard.edu/graphlan/).

## Statistical analysis

All analyses were done separately for bacterial and fungal taxa in R version 4.0.2[68]. We defined dominant taxa as those which were present across all 15 sites (management pairs) and represented the top 10% of taxa when ranked by relative abundance (rRNA reads). The response of each of these dominant taxa to drought treatment was identified using a generalized linear mixed model across all experimental plot pairs with drought treatment as a fixed effect, and region/site/field as nested random effects (R package glmmTMB version 1.1.5[69]). For each individual model, the appropriate distribution (poisson, negative binomial, or binomial) was assumed based on diagnostics of model residuals, which were assessed using R package DHARMa version 0.4.6[70]. The drought-response strategy for each taxon was identified as resistant (no significant response to drought detected), sensitive (negative response), or opportunistic (positive response) using a significance level ($\alpha$) of 0.05. Further statistical analysis (linear mixed effects models using R package nlme version 3.1–148[71]) was performed at the drought-response group level (i.e., resistant, opportunistic, sensitive, resilient, not resilient). Group-level indices were calculated as follows: for each OTU in a given group, its relative abundance in a given sample was standardized relative to its abundance across all samples; these standardized abundances were then summed across all OTUs in a given group resulting in one value (index) per group per sample.

Structural equation modelling was used to investigate effects of historical management, drought, and soil properties on relative abundances of opportunistic, sensitive, and resistant taxa at the two sampling timepoints. Within-field reps were averaged prior to analysis ($n = 180$ experimental plots/3 field replicates = 60 values per time-point). We constructed an a priori model based on current knowledge of plant-soil-microbe-functioning interactions (see Fig. S5 and Supplementary Note 1) and tested whether the data fit these models using the standard modelling approach in the lavaan R package, version 0.6-12[72]. We created a proxy for soil properties using axis 1 scores from a non-metric multidimensional scaling plot that included total soil carbon, total nitrogen, and soil temperature (see Supplementary Note 1). We used multiple parameters including root mean square error of approximation (RMSEA), comparative fit index (CFI), and Standardized Root Mean Squared Residual (SRMR) to assess model fit.

**Reporting summary**

Further information on research design is available in the Nature Portfolio Reporting Summary linked to this article.

## Data availability

The sequence data generated in this study have been deposited in the EMBL Nucleotide Sequence Database (ENA) under accession code PRJEB63076. All other data generated in this study have been deposited on GitHub[73].

## Code availability

All code is available from GitHub[73].

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

## Acknowledgements

This study was supported by a consortium grant from the Natural Environment Research Council (NERC) Soil Security Programme led by R.D.B., with component grants NE/M017028/1 to R.D.B. and F.d.V., NE/M01701X/2 to D.J. and E.M.B. and NE/M017036/1 to T.C. and M.E. We are very grateful to the landowners and farmers for allowing us to perform our experiment and sample their fields. We also thank Debbie Ashworth, Lucy Frotin, Juliette Papelard, Lupe Leon Sanchez, Heather Stott, Felix Terrell, Jake Taylor, and Tom Fenwick for help in the laboratory and field, and C. Collins for his support as coordinator of the NERC Soil Security Programme. For the purpose of open access, the author has applied a CC-BY public copyright license to any Author Accepted Manuscript version arising from this submission.

## Author contributions

R.D.B. initiated and gained funding for the project with D.J., M.E., T.C., F.d.V., and E.M.B. J.L. M.C., F.d.C., R.D.B., D.J., M.E., T.C., F.d.V. and E.M.B. conceived and designed the experiment. J.M.L., M.C., J.R., F.d.C., N.A., and M.M. set up the experiment and performed sampling and field measurements. T.G. performed amplicon sequencing and analysis. J.L., J.R., M.C., N.A., performed all other laboratory analyses. J.M.L. and M.-D.B. statistically analysed the data and led writing the manuscript in close consultation with R.D.B., T.C., D.J., and R.I.G. and with discussions and contributions from all authors.

## Competing interests

The authors declare no competing interests.
