## [Peer Review File · Nature Communications]

Reviewers' Comments:

Reviewer #1:

Remarks to the Author:

I thoroughly enjoyed reviewing this manuscript. The authors did an excellent job with the writing. It is the most well-written manuscript I have reviewed. The authors provided a generally comprehensive description of the work.

I appreciate the reasoning for the focus on dominant taxa. The takeaway that most of the dominant taxa from these soils exhibit resistant or opportunistic strategies in response to drought provides a foundation for some important theories in soil microbial ecology that have been hinted at in the literature for some time, but scarcely explicitly tested as the authors have done so here. I think this paper will help motivate other researchers in this field to make more of a point of differentiating between dominant and rare taxa and their respective roles in microbial processes and ecosystem functioning. I also appreciate the latitudinal gradient of the experimental design, which strengthens the inference power of these results and--as the authors state--make them more ecologically relevant.

I have provided minor comments in the attached word document with track changes, most of which are suggesting clarifications with phrasing or providing more detail. In particular, I would really like to see broader context added to the hypotheses and to expand on this in the Discussion. With papers like this that focus on soil microbial taxonomic composition, it's important to relay to the reader how microbial composition may influence ecosystem function. The authors have done a nice job with providing possible explanations for microbial responses to drought, land management, etc. Now they just need to bookend the story by providing some thoughts on how these microbial response strategies may affect larger scale ecosystem functions. I think the importance of this work would come through even more by making these small improvements.

**Land management shapes drought responses of dominant soil microbial taxa across**
**grasslands**

Lavallee, J. M.^{1,2}, Chomel, M.^{1,3}, Alvarez Segura, N.^{4,5}, de Castro, F.^{6,7}, Goodall, T.⁸, Magilton,
5 M.^{6,10}, Rhymes, J. M.^{1,11}, Delgado-Baquerizo, M.^{12,13}, Griffiths, R.^{8,9}, Baggs, E. M.¹⁴, Caruso, T.¹⁵,
de Vries, F. T.^{1,16}, Emmerson, M.⁶, Johnson, D.¹, Bardgett, R. D.¹

**Affiliations**

- 1. Department of Earth and Environmental Sciences, The University of Manchester, Oxford Road,
Manchester M13 9PT, UK
2. Environmental Defense Fund, 257 Park Ave S, New York, NY, 10010 USA
3. FiBL France, Research Institute of Organic Agriculture, 26400 Eurre, France
4. Institute of Biological and Environmental Sciences, University of Aberdeen, Cruickshank Building,
Aberdeen, UK
5. Department of Climate Change, EURECAT, Technological Centre of Catalonia, Spain
6. School of Biological Sciences and Institute for Global Food Security, Queen's University of Belfast,
Belfast, UK
7. AgriFood & Biosciences Institute, Belfast, UK
8. UK Centre for Ecology & Hydrology Wallingford, Maclean Building, Benson Lane, Crowmarsh
Gifford, Wallingford, Oxfordshire OX10 8BB, UK
9. School of Natural Sciences, Bangor University, Bangor, UK
10. School of Life Sciences, University of Lincoln, Lincoln, UK
11. Centre for Ecology & Hydrology Bangor, Environment Centre Wales, Deiniol Road, Bangor, UK
12. Laboratorio de Biodiversidad y Funcionamiento Ecosistémico. Instituto de Recursos Naturales y
Agrobiología de Sevilla (IRNAS), CSIC, Av. Reina Mercedes 10, E-41012, Sevilla, Spain
13. Unidad Asociada CSIC-UPO (BioFun). Universidad Pablo de Olavide, 41013 Sevilla, Spain
14. Global Academy of Agriculture and Food Systems, Royal (Dick) School of Veterinary Studies,
University of Edinburgh, Midlothian, UK
15. School of Biology and Environmental Science, University College Dublin, Dublin, Ireland
16. Institute for Biodiversity and Ecosystem Dynamics, University of Amsterdam, the Netherlands

* Corresponding Author: Jocelyn M. Lavallee, jlavallee@edf.org

Keywords: grassland, drought, bacteria, fungi, land management, soil ecology

Commented [BKK1]: Could you make this more descriptive by adding a colon? I want to know HOW drought shaped microbial responses.

**ABSTRACT**

Soil microbial communities are dominated by a relatively small number of taxa that are
ubiquitous and highly abundant relative to other taxa. Dominant taxa may play outsized roles in
ecosystem functioning, yet little is known about their capacities to resist and recover from
increasingly common climate extremes such as drought, or how environmental context mediates
those responses. Here, we imposed an *in situ* experimental drought across 30 UK grassland sites
representing a wide range of soil and climatic conditions and contrasting management intensities
and measured the responses of dominant bacteria and fungi immediately after drought and after a
post-drought recovery period. We found that the majority of dominant bacterial (85 %) and
fungal (89 %) taxa exhibited resistant or opportunistic drought strategies, which likely
contributes to their ubiquity and dominance across sites, while small proportions of dominant
microbial taxa displayed sensitive strategies in response to drought. Intensive grassland
management increased the proportion of dominant bacterial taxa with resistant and opportunistic
strategies – likely via alleviation of nutrient limitation and pH-related stress under fertilisation
and liming – but had the opposite impact on dominant fungal taxa. Establishing these links
between grassland management and drought responses of dominant soil microbial taxa is a key
step towards better predicting grassland ecosystem responses to drought.

**INTRODUCTION**

Soil microbial communities mediate ecosystem functions including nutrient cycling,
organic matter decomposition, and pathogen control¹⁻³, but their functioning can be impacted by
climate extremes^{4,5} which are becoming increasingly common. Recent evidence shows that
despite very high diversity of soil microbial taxa, a small proportion can be considered dominant,
*i.e.*, they are found across most soils and are highly abundant relative to other taxa^{6,7}. These
dominant taxa may be drivers of ecosystem responses to climate extremes (*i.e.*, the mass-ratio
hypothesis; Grime, 1998; de Vries et al., 2018), an idea supported by studies of plant
communities linking ecosystem responses to the abundances of dominant plant species^{10,11}.
Therefore, understanding how dominant microbial taxa respond to climate extremes and how
these responses are shaped by environmental factors and land management, will enable better
predictions of ecosystem behaviour into the future^{12,13}.

Soil microbial taxa can be categorised by life history strategies^{14,15} to inform on their
capacity to resist and recover from climate extremes such as drought^{12,16}. These life history
strategies are thought to emerge from correlated sets of traits (*e.g.*, related to resource
acquisition, growth yield, and stress tolerance), which are favoured under different
environmental conditions¹⁵. For example, soil microbial communities subjected to moisture
pulses had greater proportions of taxa exhibiting a stress-resistant strategy, whereas those under
ambient conditions had higher abundances of drought-sensitive taxa¹⁴. Land management may
also shift microbial life history strategies by changing resource availability and plant
communities – environmental factors known to shape microbial community structure and
function^{17–22}. However, the interacting effects of land management and climate extremes such as
drought have not been studied in the context of microbial life history strategies. This is a
necessary step towards using ecological knowledge of soil microbes to predict and understand
the consequences of land management decisions on soil functioning and sustainability in the face
of climate change.

Here, we carried out a large-scale field experiment across a broad range of grassland sites
to explore how the relative abundances of dominant microbial taxa with different drought-
response strategies are shaped by soil conditions, climate, and land management intensity. We
imposed a simulated drought on 15 pairs of grasslands under contrasting management (*i.e.*,
intensive and extensive) in three geographically distinct regions of the UK representing a range
of soil and climatic conditions (Fig. S1, Table S1). Using an operational approach, we identified
dominant microbial taxa and classified them into three broad drought-response strategies (*i.e.*,
opportunistic, Wean, and resistant) based on their abundance and growth rates in plots
immediately following the drought and after a 60-day post-drought period²³ – to capture
microbial resistance and resilience to drought²⁴.

We hypothesised that: (1) dominant soil microbial taxa largely display resistant or
opportunistic strategies under drought, because a capacity to withstand variable moisture
conditions would partly explain their ubiquity and abundance across sites; (2) intensive grassland
management, characterized by regular fertiliser and lime application and higher plant
productivity (Table S1), favours taxa that are maladapted to low resource availability and stress,
and therefore will be sensitive to drought; and (3) intensive grassland management favours

Commented [BKK2]: Even though you do so in the Methods, it would be helpful to briefly define the strategies here. I think it's also worth explicitly differentiating between resistant and resilient because many people outside of this literature often use them interchangeably.

Commented [BKK3]: It would be helpful to add broader context to the hypotheses. What would each of these hypotheses, or them collectively, indicate about microbial function under climate extremes if they were supported?

microbial taxa that recover after drought (*i.e.*, resilient), because more favourable soil conditions
allow drought-affected taxa to rebound quickly with rewetting.

RESULTS

We found that a relatively small number of bacteria and fungi dominate soils across the
grassland sites, and that these taxa were highly resistant to an imposed drought event. For
bacteria, dominant taxa (defined as present across all 15 sites and in the top 10 % of relative
abundance ranked by 16S rRNA reads⁷) represented 1269 out of 19224 total operational
taxonomic units (OTUs), which accounted for approximately 7 % of total OTUs but 76 % of all
reads. For fungi, dominant taxa (present across all three regions and in the top 10 % by ITS
rRNA reads) made up 209 out of 12837 total OTUs, accounting for approximately 2 % of total
OTUs but 53 % of all reads. Overall, the majority of dominant bacterial (66 %) and fungal (64
109 %) taxa were classified as displaying a resistant drought strategy, as they showed no response to
110 drought in our hierarchical model using all data across sites and management regimes
immediately after the simulated drought (Table S2). Opportunistic taxa, whose relative
abundances increased in response to drought, represented 19 % of dominant bacteria and 25 % of
dominant fungi; sensitive taxa, whose relative abundances decreased with drought, represented
12 % of dominant bacteria and 7 % of dominant fungi.

**Figure 1.** Dominant bacterial (a) and fungal (b) community responses to drought immediately
following drought (“resistance”) and after the 60-day post-drought period (“resilience”), limited
to the top 500 most abundant taxa across all samples for readability. The inner ring shows the

Commented [BKK4]: It would be helpful to have bacteria and fungi labels at the top or bottom of these trees so that it's immediately obvious to the reader which is which.

Also, I don't recall any Methods text that describes how the trees were made.

taxonomic tree, coloured by phylum. The middle ring displays drought response strategies
(“resistance”, immediately following drought) of each OTU (light blue = ~~tolerant~~ resistant, black
= opportunistic, pink = sensitive). The outer ring displays responses after the 60-day post drought
period (“resilience”; green = resilient taxa that recovered to control levels, dark grey = not
resilient). Taxa defined as resistant to the drought (light blue, inner ring) were not tested for
resilience. For further information on the identities of all OTUs, see Supplemental Note 2.

Dominant bacterial phyla in our dataset comprised primarily (by reads) *Proteobacteria*
(32 %), *Acidobacteria* (21 %), *Verrucomicrobia* (13 %), *Bacteroidetes* (11 %), *Firmicutes* (9 %),
*Actinobacteria* (7 %), *Chloroflexi* (3 %), and several other globally distributed taxa. Of these
phyla, most contained taxa representing each of the three drought-response strategies (Fig. 1).
However, members of *Firmicutes* and *Bacteroidetes* tended to display resistant or sensitive
drought-response strategies, with few or no taxa identified as opportunistic (zero out of 47 in
*Firmicutes*; five out of 175 in *Bacteroidetes*). Members of *Acidobacteria*, *Actinobacteria*, and
*Chloroflexi* tended to display resistant or opportunistic drought-response strategies, with few taxa
identified as sensitive (nine out of 227 in *Acidobacteria*, one out of 118 in *Actinobacteria*, and
one out of 65 in *Chloroflexi*). Dominant fungal phyla comprised (by reads) *Mortierellomycota*
(48 %), *Ascomycota* (22 %), *Basidiomycota* (15 %), *Glomeromycota* (1 %), and several other
known and globally distributed or unidentifiable taxa. Members of *Ascomycota* tended to display
resistant or opportunistic drought-responses strategies, with only six of 94 taxa identified as
having a drought-sensitive strategy. Members of *Mortierellomycota*, *Basidiomycota*, and
*Glomeromycota* tended to display resistant or sensitive drought-response strategies, with only
one or no taxa identified as opportunistic in each phylum (Fig. 1). Overall, dominant taxa
resistant to drought belonged to different taxonomic groups dispersed across every major lineage
of the phylogeny, suggesting that this capability is not limited to specific phylogenetic groups of
microbes.

Commented [BKK5]: I would appreciate an Excel table of all dominant taxa, their full taxonomy, and response strategy. It's fine to not provide all this detail in the manuscript, but I think it should be made available. I'm curious (and I'm sure many other readers will be, too) to know which orders, families, etc. have resistant and sensitive taxa.

[revised manuscript text omitted]

Commented [BKK6]: It would helpful to add in parentheses a brief description of oligotrophs as they relate to copiotrophs. Ex: generally have a slow growth strategy and thrive in nutrient-poor conditions

here that intensive management reduces the flux of recent photosynthate to soil food webs
including arbuscular mycorrhizal fungi, indicating importance of this pathway for driving fungal
activity⁴⁰. It is therefore possible that this pathway of reduced energy flux could contribute to
the increased sensitivity of dominant fungal communities to drought (which further reduces the
flux of recent photosynthate below-ground⁴¹) in intensively managed grasslands. The opposing
responses of dominant bacteria and fungi to grassland management in terms of their resistance
and resilience to drought may help to explain widespread observations of increasing
bacterial:fungal biomass ratios with grassland intensification^{18,39,42}.

Overall, the alignment of resistance and resilience in the context of grassland
management intensity for both bacteria and fungi was unexpected, as other studies have found
observed trade-offs between resistance and resilience in soil microbial communities⁴³⁻⁴⁵.
However, in our study encompassing a relatively broad range of soils, pH was an important
driver of both resistance and resilience in dominant bacterial communities, while fertilisation
may have driven both resistance and resilience of dominant fungal communities, which would
help to explain the alignment in both responses with management. While consistencies in taxon-
level responses to separate drought and nitrogen addition treatments has been observed
previously⁴⁶, the sets of traits determining responses to soil water availability versus nutrient
availability or pH may not always align and a multi-dimensional framework may be necessary
for considering microbial life history strategies⁴⁷ and predicting microbial responses to climate
extremes.

We observed contrasting phylum-level responses to drought in soil dominant bacterial
and fungal communities, suggesting that certain phyla may be inherently more resistant and
resilient to drought than others. *Actinobacteria* contained a high proportion of resistant and
opportunistic taxa, with only one taxon identified as sensitive, consistent with previous
observations that *Actinobacteria* are prevalent in dry environments⁴⁸ and are highly resistant or
increase in response to drought^{4,47}. Members of *Firmicutes* and *Bacteroidetes* were generally
more sensitive to the drought treatment, and while members of *Bacteroidetes* have been shown
to decrease in relative abundance in drier soils, members of *Firmicutes* have previously shown
the opposite response⁴⁹. It is possible that these previous observations may have been driven
primarily by one or a few taxa, which may not have been present (or defined as dominant) here.
Context-dependent drought responses have been previously observed for other phyla including

Commented [BKK7]: These ratios are usually expressed as fungi:bacteria.

*Proteobacteria* and *Planctomycetes*⁴⁹, and we also observed relatively high numbers of taxa
with different drought-response strategies in those phyla.

Dominant members of *Ascomycota* were particularly opportunistic under drought, which
agrees with findings that *Ascomycota* are dominant globally and are generalists that are adapted
to a wide range of conditions^{6,48}. Within *Glomeromycota*, dominant taxa that responded to our
drought treatment were sensitive, in agreement previous findings that both community
composition⁴ and functionality^{50,51} of this group of fungi respond to drought in other systems.
However, the majority of dominant *Glomeromycota* in this study were found to be resistant to
the drought, suggesting that the results from these other studies may largely be driven by only a
few dominant members of *Glomeromycota*, or by taxa that were not defined as dominant here.
Two members of *Basidiomycota* indicated sensitivity to drought, and one taxon did not recover
after the 60-day post-drought period. Given that *Basidiomycota* are important decomposers and
ectomycorrhizal symbionts in forests⁵², microbial communities in forested systems (or under
forest expansion) may be sensitive to drought with potential implications for forest growth and
ecosystem functioning⁵³, which deserves further study.

Overall, our findings from a broad range of grassland sites across the UK indicate that
~~most the majority~~ of ~~the~~ dominant soil microbial taxa are highly resistant to drought, which may
explain their prevalence across a diverse range of grassland soils. We further show that grassland
management, along with climate and soil properties, shapes the relative abundances of dominant
soil microbial taxa with differing drought-response strategies. More intensive grassland
management, which creates more optimal pH and higher nitrogen availability compared to
extensive management, promoted opportunistic and resilient bacterial taxa that may employ
copiotrophic or fast-response strategies and are able to take advantage of changing conditions.
However, it has the opposite effect on dominant fungal taxa which may help to explain increases
in bacterial prevalence over fungi with grassland intensification^{18,39,42,54}. By demonstrating that
land management shapes the drought-response strategies of dominant microbial taxa across
grasslands, our findings improve our understanding of how soil microbial communities respond
to drought. ~~Moreover, by identifying consistent management- and drought-induced responses of~~
~~dominant microbial taxa, our findings pave the way for future studies that interrogate their~~
~~functional attributes and links to key ecosystem functions~~⁵⁵. ~~Given the enormous complexity of~~
soil microbial communities and their dynamics in space and time, our approach of focusing on

Commented [BKK8]: You've done a great job with the discussion, and the cherry on top would be to expand on this sentence a bit more—like I suggested in the hypotheses section of the Introduction.

Just providing a few sentences about how these microbial strategy responses to drought and land management may impact particular ecosystem functions like carbon, nutrient cycling, etc. This would really help drive home the importance of this work.

the drought response strategies of dominant taxa is one way to make this task more feasible in
the future.

**METHODS**

*Field sites*

[revised manuscript text omitted]

Commented [BKK9]: Do you mean organic matter? If so, just spell it out.

GCTATTGGAGCTGGAATTAC, respectively. Amplicons were generated using high-fidelity
DNA polymerase Q5 Taq (New England Biolabs, Ipswich, USA). After an initial denaturation at
95 °C for 2 minutes, PCR conditions were: denaturation at 95 °C for 15 seconds, annealing at 55
447 °C for 30 seconds with extension at 72 °C for 30 seconds, repeated for 30 cycles, followed by a
448 final extension of 10 minutes at 72 °C.

449 Fungal internal transcribed spacer (ITS) amplicon sequences were generated using a 2-step
amplification approach. Primers GTGARTCATCGAATCTTTG and
TCCTCCGCTTATTGATATGC⁶¹ were each modified at the 5' end with the addition of
Illumina pre-adapter and Nextera sequencing primer sequences. After an initial denaturation at
95°C for 2 minutes, PCR conditions were: denaturation at 95°C for 15 seconds, annealing at
52°C for 30 seconds with extension at 72°C for 30 seconds, repeated for 25 cycles, with a final
extension of 10 minutes at 72°C included.

Sequenced paired-end reads were joined using PEAR (Zhang et al., 2014), quality filtered
using FASTX tools (hannonlab.cshl.edu), and length-filtered to a minimum length of 300 bp.
The presence of PhiX and adaptors were checked for and removed with BBTools
(jgi.doe.gov/data-and- tools/bbtools/), and chimeras were identified and removed with
VSEARCH_UCHIME_REF⁶² using Greengenes Release 13_5 (at 97%). Singletons were
removed and the resulting sequences were clustered into operational taxonomic units (OTUs)
with VSEARCH_CLUSTER⁶² at 97% sequence identity. Representative sequences for each
OTU were taxonomically assigned by RDP Classifier with the bootstrap threshold of 0.8 or
greater using the Greengenes Release 13_5 (full) as the reference. Unless stated otherwise,
default parameters were used for all steps listed. The fungal ITS sequences were analysed using
PIPITS⁶³ with default parameters. Briefly, this involved quality filtering and 97% clustering of
the ITS2 region as indicated above for the 16S processing, using the UNITE database for
chimera removal and taxonomic identification of representative OTUs. Both bacterial and fungal
OTU abundance tables were resampled to a minimum of 9000 reads per sample, and samples
with zero reads were removed prior to further analyses.

*Statistical analysis*

All analyses were done separately for bacterial and fungal taxa in R version 4.0.2⁶⁴. We defined
dominant taxa as those which were present across all 15 sites (management pairs) and
represented the top 10% of taxa when ranked by relative abundance (rRNA reads). The
responses of these dominant taxa to drought treatment were tested using generalized linear mixed
models across all experimental plot pairs with drought treatment as a fixed effect, and
region/site/field as nested random effects (R package glmmTMB;⁶⁵). Poisson, negative
binomial, or binomial distributions were assumed based on diagnostics of model residuals, which
were assessed using R package DHARMA⁶⁶. The drought-response strategy for each taxon was
identified as resistant (no significant response to drought detected), sensitive (negative response),
or opportunistic (positive response) using a significance level (α) of 0.05. Prior to statistical
analysis, indices for each group (by drought-response strategy) were calculated as follows: for

Commented [BKK10]: Do you mean rarefied? If so, consider using the word rarefied.

Commented [BKK11]: Don't need the semicolon here

each OTU in a given group, its relative abundance in a given sample was standardized relative to
its abundance across all samples; these standardized abundances were then summed across all
OTUs in a given group resulting in one value per group per sample.

Structural equation modelling was used to investigate effects of historical management,
drought, and soil properties on relative abundances of opportunistic, sensitive, and resistant taxa
at the two sampling timepoints. Within-field reps were averaged prior to analysis ($n = 180/3 = 60$ per
timepoint). We constructed an *a priori* model based on current knowledge of plant-soil-microbe-
functioning interactions (see Fig. S5 and Supplemental Note 1) and tested whether the data fit
these models using the standard modelling approach in the lavaan R package⁶⁷. We created a
proxy for soil properties using axis 1 scores from a non-metric multidimensional scaling plot that
included total soil carbon, total nitrogen, and soil temperature (see Supplemental Note 1). We
used multiple parameters including root mean square error of approximation (RMSEA),
comparative fit index (CFI), and Standardized Root Mean Squared Residual (SRMR) to assess
model fit.

REFERENCES

- 1. Wagg, C., Bender, S. F., Widmer, F. & van der Heijden, M. G. A. Soil biodiversity and
soil community composition determine ecosystem multifunctionality. *Proc. Natl. Acad.*
*Sci. U. S. A.* **111**, 5266–5270 (2014).
- 2. Bardgett, R. D. & van der Putten, W. H. Belowground biodiversity and ecosystem
functioning. *Nature* **515**, 505–511 (2014).
- 3. Delgado-Baquerizo, M. *et al.* Microbial diversity drives multifunctionality in terrestrial
ecosystems. *Nat. Commun.* **7**, 1–8 (2016).
- 4. Ochoa-Hueso, R. *et al.* Drought consistently alters the composition of soil fungal and
bacterial communities in grasslands from two continents. *Glob. Chang. Biol.* **24**, 2818–
2827 (2018).
- 5. Oliverio, A. M., Bradford, M. A. & Fierer, N. Identifying the microbial taxa that
consistently respond to soil warming across time and space. *Glob. Chang. Biol.* **23**, 2117–
2129 (2017).
- 6. Egidi, E. *et al.* A few Ascomycota taxa dominate soil fungal communities worldwide. *Nat.*
*Commun.* **10**, (2019).
- 7. Delgado-Baquerizo, M. *et al.* A global atlas of the dominant bacteria found in soil.
*Science (80-.)*. **359**, 320–325 (2018).
- 8. Grime, J. P. Benefits of plant diversity to ecosystems: immediate, filter and founder
effects. *J. Ecol.* **86**, 902–910 (1998).
- 9. de Vries, F. T. *et al.* Soil bacterial networks are less stable under drought than fungal
networks. *Nat. Commun.* **9**, 912–913 (2018).
- 10. Hoover, D. L., Knapp, A. K. & Smith, M. D. Resistance and resilience of a grassland
ecosystem to climate extremes. *Ecology* **95**, 2646–2656 (2014).
- 11. Smith, M. D. *et al.* Mass ratio effects underlie ecosystem responses to environmental
change. *J. Ecol.* **108**, 855–864 (2020).
- 12. Bardgett, R. D. & Caruso, T. Soil microbial community responses to climate extremes:
Resistance, resilience and transitions to alternative states. *Philos. Trans. R. Soc. B Biol.*
*Sci.* **375**, (2020).

Commented [BKK12]: For clarification: were separate models run for each “dominant” OTU? I’m confused with the grouping described here. What is meant by “indices for each group”?

[revised manuscript text omitted]

[project.org/](http://www.r-project.org/) (2020).
- 65. Mollie E. Brooks *et al.* glmmTMB Balances Speed and Flexibility Among Packages for
Zero-inflated Generalized Linear Mixed Modeling. *R J.* **9**, 378–400 (2017).
- 66. Hartig, F. DHARMA: Residual Diagnostics for Hierarchical (Multi-Level / Mixed)
Regression Models. at <https://cran.r-project.org/package=DHARMA> (2022).
- 67. Rosseel, Y. lavaan: An R Package for Structural Equation Modeling. *J. Stat. Softw.* **48**, 1–

36 (2012).

**ACKNOWLEDGEMENTS**

This study was supported by a consortium grant from the Natural Environment Research Council
(NERC) Soil Security Programme led by R.D.B., with component grants NE/M017028/1 to
R.D.B. and F.d.V., NE/ M01701X/2 to D.J. and E.M.B. and NE/M017036/1 to T.C. and M.E.

We are very grateful to the landowners and farmers for allowing us to perform our experiment
and sample their fields. We also thank Debbie Ashworth, Lucy Frotin, Juliette Papeard, Lupe
Leon Sanchez, Heather Stott, Felix Terrell, Jake Taylor, and Tom Fenwick for help in the
laboratory and field, and C. Collins for his support as coordinator of the NERC Soil Security
Programme. For the purpose of open access, the author has applied a CC-BY public copyright
license to any Author Accepted Manuscript version arising from this submission.

**AUTHOR CONTRIBUTIONS**

R.D.B. initiated and gained funding for the project with D.J., M.E., T.C., F.d.V., and E.B.. J.L.
683 M.C., F.d.C., R.D.B., D.J., M.E., T.C., F.d.V. and E.B. conceived and designed the experiment.
684 J.L., M.C., J.R., F.d.C., N.A., and M.M. set up the experiment and performed sampling and field
measurements. T.G. performed amplicon sequencing and analysis. J.L., J.R., M.C., N.A.,
performed all other laboratory analyses. J.L. and M.D.B. statistically analysed the data and led
writing the manuscript in close consultation with R.D.B., T.C., D.J., and R.G. and with
discussions and contributions from all authors.

**SUPPLEMENTAL INFORMATION**

**Land management shapes drought responses of dominant soil microbial taxa across grasslands**

Lavallee, J. M.^{1,2}, Chomel, M.^{1,3}, Alvarez Segura, N.^{4,5}, de Castro, F.^{6,7}, Goodall, T.⁸, Magilton, M.^{6,10}, Rhymes, J. M.^{1,11}, Delgado-
Baquerizo, M.^{12,13}, Griffiths, R.^{8,9}, Baggs, E. M.¹⁴, Caruso, T.¹⁵, de Vries, F. T.^{1,16}, Emmerson, M.⁶, Johnson, D.¹, Bardgett, R. D.¹.

**Affiliations**

1. Department of Earth and Environmental Sciences, The University of Manchester, Oxford Road, Manchester M13 9PT, UK

2. Environmental Defense Fund, 257 Park Ave S, New York, NY, 10010 USA

3. FiBL France, Research Institute of Organic Agriculture, 26400 Eurre, France

4. Institute of Biological and Environmental Sciences, University of Aberdeen, Cruickshank Building, Aberdeen, UK

5. Department of Climate Change, EURECAT, Technological Centre of Catalonia, Spain

6. School of Biological Sciences and Institute for Global Food Security, Queen's University of Belfast, Belfast, UK

7. AgriFood & Biosciences Institute, Belfast, UK

8. UK Centre for Ecology & Hydrology Wallingford, Maclean Building, Benson Lane, Crowmarsh Gifford, Wallingford, Oxfordshire OX10
8BB, UK

9. School of Natural Sciences, Bangor University, Bangor, UK

10. School of Life Sciences, University of Lincoln, Lincoln, UK

11. Centre for Ecology & Hydrology Bangor, Environment Centre Wales, Deiniol Road, Bangor, UK

12. Laboratorio de Biodiversidad y Funcionamiento Ecosistémico. Instituto de Recursos Naturales y Agrobiología de Sevilla (IRNAS), CSIC, Av.
Reina Mercedes 10, E-41012, Sevilla, Spain

13. Unidad Asociada CSIC-UPO (BioFun). Universidad Pablo de Olavide, 41013 Sevilla, Spain

14. Global Academy of Agriculture and Food Systems, Royal (Dick) School of Veterinary Studies, University of Edinburgh, Midlothian, UK

15. School of Biology and Environmental Science, University College Dublin, Dublin, Ireland

16. Institute for Biodiversity and Ecosystem Dynamics, University of Amsterdam, the Netherlands

* Corresponding Author: Jocelyn M. Lavallee, jlavallee@edf.org

**Figure S1.** Map showing site locations, with detail for each of the three regions (Devon, North Yorkshire, Aberdeenshire). Each point

represents one site consisting of two paired fields with contrasting management (extensive, intensive).

**Table S1.** Characteristics of the grassland sites used in this study. Each site consists of paired pastures under contrasting management
 (“extensive” or “intensive”). Paired of fields at each site are adjacent or located < 0.5 km apart.

Region	Site #	Intensive					Extensive				
		Fertilisation	Lime application	Cuttings	Number of plant species	Soil bulk density	Fertilisation	Lime application	Cuttings	Number of plant species	Soil bulk density
Devon	1	Synthetic NPK	Yes	1-2 times per year	≥ 5	0.65	Unfertilized for ≥ 15 years	None	None	≥ 9	0.53
	2	Synthetic NPK and manure slurry	3 tonnes acre ⁻¹ applied in 2010	2 times per year	≥ 5	0.65	Unfertilized for ≥ 50 years	None	None	≥ 15	0.61
	3	Synthetic NPK	Yes	Once per year	≥ 5	0.94	Unfertilized for ≥ 20 years	None	Once per year	≥ 13	0.88
	4	Synthetic NPK (100 kg 20-10-10 acre ⁻¹ year ⁻¹)	Yes	1-2 times per year	≥ 7	0.76	Unfertilized for ≥ 20 years	None	None	≥ 13	0.63
	5	Manure slurry	2 tonnes acre ⁻¹ applied in 2009	3 times per year	≥ 6	0.84	Unfertilized for ≥ 10 years	None	None	≥ 9	0.61
North Yorkshire	1	Synthetic NPK (50 kg 25-5-5 acre ⁻¹ year ⁻¹) and manure	Yes	Not in past 17 years	≥ 9	0.60	Unfertilized for ≥ 15 years	None	None	≥ 9	0.68
	2	Synthetic NPK and manure	1 tonne acre ⁻¹ every 3 years	Once per year	≥ 9	0.34	Unfertilized for ≥ 65 years	None	None	≥ 15	0.31
	3	Synthetic NPK (50 kg 25-5-5 acre ⁻¹ year ⁻¹) and manure	2 tonnes acre ⁻¹ every 7 years	Once per year	≥ 11	0.44	Unfertilized for ≥ 50 years	None	None	≥ 11	0.60
	4	Synthetic NPK and manure	Yes	Once per year	≥ 8	0.33	Unfertilized for ≥ 10 years	None	None	≥ 10	0.65
	5	Synthetic NPK (75 kg 25-5-5 acre ⁻¹	Once per year	Once per year	≥ 8	0.53	Unfertilized for ≥ 17 years	None	None	≥ 9	0.18

		year ⁻¹) and manure									
Aberdeen-shire	1	Synthetic NPK (75 kg 24-0-0 acre ⁻¹ year ⁻¹ , P-K some years)	1.6 tonnes acre ⁻¹ in 2012	Once per year	≥ 10	0.61	Unfertilized for ≥ 30 years	None	None	≥ 15	0.50
	2	Synthetic NPK (100 kg 25-5-5 acre ⁻¹ year ⁻¹)	2 tonnes acre ⁻¹ every 6 years	N/A	≥ 7	0.85	Unfertilized for ≥ 25 years	None	None	≥ 13	0.74
	3	Synthetic NPK (100 kg 25-5-5 acre ⁻¹ year ⁻¹) and manure	2 tonnes acre ⁻¹ every 6 years	N/A	≥ 6	0.75	Unfertilized for ≥ 25 years	None	None	≥ 7	0.72
	4	Synthetic NPK or P-K until 2013	1.62 tonnes acre ⁻¹ in 2011	N/A	≥ 4	0.83	Unfertilized for ≥ 25 years	None	None	≥ 7	0.61
	5	Synthetic NPK or P-K until 2009	0.8 tonnes acre ⁻¹ in 2005	N/A	≥ 6	0.74	Unfertilized for ≥ 25 years	None	None	≥ 8	0.62

Figure S2. Effects of drought and grassland management on soil moisture (a), ecosystem respiration (b), aboveground plant biomass (c), and microbial biomass (d), measured immediately following removal of drought shelters. P values from linear mixed effect models for drought (Dr), management (Ma), and their interactions (Dr*Ma) are given.

**Figure S3.** Constantly-monitored moisture probe data for one drought-control pair in each of the three regions shows reductions in
 soil moisture for the duration of the experimental drought period.

**Table S2.** Count and % of dominant bacterial (16S) and fungal (ITS) OTUs classified under each drought response strategy across all
 sites. Values in parenthesis are the number or percentage (of total) of opportunistic or sensitive OTUs classified as resilient.

Drought response strategy	Bacteria		Fungi	
	Count	% of dominant OTUs	Count	% of dominant OTUs
Resistant	841	66.3	134	64.1
Opportunistic	243 (192)	19.1 (15.1)	52 (35)	24.8 (16.7)
Sensitive	185 (151)	14.6 (11.9)	23 (15)	11 (7.2)
Total	1269		209	

**Table S3.** Results of perMANOVA analysis of Bray-Curtis dissimilarities using all relative abundance data for a) bacterial (16S) and
 b) fungal (ITS) genes in relation to drought treatment, region, management and their interactions. Df = degrees of freedom;
 SumsOfSqs = sums of squares; F.Model = F value by permutation; p values based on 999 permutations (lowest p-value possible is
 0.001). Asterisks indicate significance at $\alpha = 0.05$. Due to the structure of the model call (stratification at field level), terms that do not
 include drought (gray text) are not accurately represented and should be ignored.

a)	Df	SumsOfSqs	MeanSqs	F.Model	R ²	Pr(>F)	
Region	2	3.445	1.72238	23.3038	0.10548	0.001	***
Management	1	2.27	2.27045	30.7192	0.06952	0.001	***
Drought	1	0.271	0.27108	3.6677	0.0083	0.001	***
Field	26	15.865	0.61018	8.2557	0.48578	0.001	***
Region:Drought	2	0.199	0.09935	1.3441	0.00608	0.001	***
Management:Drought	1	0.095	0.09454	1.2791	0.00289	0.019	*
Region:Management:Drought	2	0.167	0.08333	1.1274	0.0051	0.054	.
Residuals	140	10.347	0.07391	0.31684			
Total	175	32.658	1				

b)	Df	SumsOfSqs	MeanSqs	F.Model	R ²	Pr(>F)	
Region	2	7.644	3.822	28.3605	0.14127	0.001	***
Management	1	2.492	2.4919	18.4903	0.04605	0.001	***
Drought	1	0.537	0.5367	3.9824	0.00992	0.001	***
Field	26	24.16	0.9292	6.8952	0.44652	0.001	***
Region:Drought	2	0.355	0.1774	1.3163	0.00656	0.014	*
Management:Drought	1	0.182	0.182	1.3504	0.00336	0.033	*
Region:Management:Drought	2	0.275	0.1377	1.022	0.00509	0.4	
Residuals	137	18.463	0.1348	0.34122			
Total	172	54.108	1				

60

61

Figure S4. Relationships between soil pH and standardized relative abundances of resistant (a) and resilient (b) bacterial taxa and resistant (c) and resilient (d) fungal taxa across all samples collected in this experiment. Lines are polynomial fits of the data produced with the ggplot2 package in R, with corresponding fit equations and multiple R^2 values.

**Supplemental Note 1: Structural Equation Models (SEMs)**

*Variables*

Microbial drought response groupings (“indices” in the main text) in the SEMs are sums of the relative abundances of each OTU
standardized relative to their abundances across all samples for each drought response category (opportunistic, sensitive, tolerant).

The “Soil” variable in the SEMs was created by extracting sample values for the first axis of a non-metric multidimensional scaling
(NMDS) plot that included total soil carbon, total soil nitrogen, soil texture (% sand), and soil temperature measured at each sampling
event. Prior to creating the NMDS, we used a principal coordinates of neighbourhood matrix (PCNM) to account for spatial
autocorrelation in the data; spatial autocorrelation accounted for about 16% of spatial variation, while latitude accounted for about
19%. We used the residuals from the spatial model that included the important PCNM vectors (but not latitude) to build the NMDS,
and we modeled latitudinal effects explicitly in the SEM.

Grassland management was represented as a binary variable (0 = extensive, 1 = intensive). Drought treatment was represented in the
same way (0 = control, 1 = drought).

*Hypothesized paths*

We expected latitude to affect ecosystem properties (soil C and N, soil temperature, pH, soil water content) due to its correlation with
climate, a primary control on soil and plant properties. We expected soil properties to influence soil water content via effects of
organic matter content on soil water holding capacity¹.

We expected grassland management to affect soil C, and N (components of the “Soil” variable) by influencing plant productivity
(inputs to soil C) and nutrient content through fertilization, liming, haying, and seeding. We also expected management to influence
pH through liming and fertilization², and soil water content via multiple mechanisms (e.g., moisture dynamics via plant biomass and
rooting structure).

We expected the relative abundances (normalized as indices) of each group of dominant microbial taxa to be affected by variables that
have been shown to control microbial community structure: (1) latitude due to its correlation with climate, (2) soil properties (organic
matter content (C and N)³⁻⁵ and temperature, (3) pH^{6,7}, (4) soil water content⁸. We expected the drought treatment to impact microbial
groups separately from soil water content because soil water content was only measured at the individual sampling points, while the

Commented [BKK1]: I'm curious as to why pH and SWC were not also grouped in with C, N, texture, and soil temp or why each was not its own factor in the SEMs?

I don't have much experience with running SEMs myself, but are you unintentionally overweighting pH and SWC in respect to the other edaphic variables by grouping them this way?

drought treatment variable was a broader representation of the treatment itself (potentially reflecting effects aside from soil water
content on those particular days). We hypothesized that intensive management would affect dominant microbial groups via the
mechanisms described in the main text.

Figure S5. *A priori* Structural Equation Model testing the relationships between grassland management, drought treatment, latitude, soil properties, soil moisture, and dominant microbial taxa (grouped according to drought response strategy). See Supplementary Note 1 for additional explanations of variables and paths.

**Supplemental Note 2: OTU Information**

Sequences will be submitted to the European Bioinformatics Institute (<https://www.ebi.ac.uk/>) during the review process.

**Supplemental Methods**

*PLFA extraction and analysis*

Soil microbial communities were characterised by the extraction of the phospholipid fatty acids (PLFAs), according to Buyer and
Sasser⁹ and detailed in Chomel et al. ¹⁰. The quantities of individual PLFAs were determined by gas chromatography using an Agilent
Technologies 7890B gas chromatograph with an Agilent DB-5 ms column. The internal standard 19:0 phosphati-dylcholine (Avanti
Polar Lipids, Inc.; Birmingham, AL, USA) added at the beginning of the extraction procedure was used for calculating concentrations.
The fatty acids i15:0, a15:0, 15:0, i16:0, 16:1 ω 7, i17:0, a17:10, 17:0, cy17:0, 18:1 ω 7, and cy19:0 were used as bacterial markers while
the fatty acid 18:2 ω 6,9 was used as a marker of fungi (Fostergard and Baath, 1996; de Vries et al., 2018). Microbial biomass C was
calculated assuming 1mg C is equal to 363.6 nmol ¹¹ bacterial PLFA or 11.8 nmol fungal PLFA ¹², and total microbial biomass was
calculated as the sum of bacterial and fungal biomass.

*Plant-available Nitrogen*

Plant-available N was extracted using 5 g field-moist, homogenised soil mixed with 25 mL of 1 M KCl in a 50 mL centrifuge tubes
which were shaken horizontally for 1 hour, then left to stand until settled. Extracts were passed through Whatman No. 1 filter paper
and analysed colorimetrically on an autoanalyser (AA3 HR AutoAnalyser, Seal Analytical, Soton, UK).

*Ecosystem respiration and net ecosystem exchange (NEE)*

Gas sampling was performed at each time point (days 0, 8, 20, and 60) to assess ecosystem respiration and net ecosystem exchange
(NEE). Gas sampling collars were installed to 6 cm depth (15 cm diameter) in each plot at the same time as the drought treatments
were established. All gas measurements were made between 10 am and 2 pm – with a maximum of 20 minutes between measurements
of paired fields – using a closed loop technique with a custom chamber (2.3 L) attached to an EGM-4 (PP Systems, Amesbury, MA,
USA). For NEE, a transparent chamber was placed on top of the gas sampling collar (sealed with a rubber gasket) and the
concentration of CO₂ in the chamber headspace was measured for two minutes. After removing the chamber to vent, respiration was
measured in the same way with an opaque cover placed over the chamber to block all incoming light. For all gas measurements, the
first 24 s of data were ignored to account for chamber equilibration and the remaining data was assessed for linearity ($R^2 > 0.7$), with a
minimum of 60 s used to calculate an average flux rate.

*Aboveground plant biomass*

At day 0 only, aboveground plant biomass was collected to assess the impact of the drought on the plant community. In each of the
180 experimental plots, a small quadrat was randomly placed with no edge less than 15 cm inside the plot boundary and all
aboveground plant biomass inside the quadrat was cut to within 4 cm of the soil surface. Biomass was collected in paper bags and
transported to the laboratory within 24-28 hours where it was dried to constant weight at 40°C, weighed, and stored for further
analysis.

**Supplemental References**

- 1. Cotrufo, M. F. & Lavellee, J. M. Soil organic matter formation, persistence, and functioning: A synthesis of current
understanding to inform its conservation and regeneration. in *Advances in Agronomy* 1–66 (Elsevier Inc., 2021).
doi:10.1016/bs.agron.2021.11.002.
- 2. Tian, D. & Niu, S. A global analysis of soil acidification caused by nitrogen addition. *Environ. Res. Lett.* **10**, 24011–24019
(2015).
- 3. Ramirez, K. S., Craine, J. M. & Fierer, N. Consistent effects of nitrogen amendments on soil microbial communities and
processes across biomes. *Glob. Chang. Biol.* **18**, 1918–1927 (2012).
- 4. Fierer, N., Bradford, M. A. & Jackson, R. B. Toward an ecological classification of soil bacteria. *Ecology* **88**, 1354–1364
(2007).
- 5. Rousk, J. *et al.* Soil bacterial and fungal communities across a pH gradient in an arable soil. *ISME J.* **4**, 1340–1351 (2010).
- 6. Delgado-Baquerizo, M. *et al.* A global atlas of the dominant bacteria found in soil. *Science (80-.)*. **359**, 320–325 (2018).
- 7. Lauber, C. L., Strickland, M. S., Bradford, M. A. & Fierer, N. The influence of soil properties on the structure of bacterial and
fungal communities across land-use types. *Soil Biol. Biochem.* **40**, 2407–2415 (2008).
- 8. Manzoni, S., Schimel, J. P. & Porporato, A. Responses of soil microbial communities to water stress: Results from a meta-
analysis. *Ecology* **93**, 930–938 (2012).
- 9. Buyer, J. S. & Sasser, M. High throughput phospholipid fatty acid analysis of soils. *Appl. Soil Ecol.* **61**, 127–130 (2012).
- 10. Chomel, M. *et al.* Drought decreases incorporation of recent plant photosynthate into soil food webs regardless of their trophic
complexity. *Glob. Chang. Biol.* **25**, 3549–3561 (2019).
- 11. Frostegård, A. & Bååth, E. The use of phospholipid fatty acid analysis to estimate bacterial and fungal biomass in soil. *Biol.*
*Fertil. Soils* **22**, 59–65 (1996).
- 12. Klamer, M. & Bååth, E. Estimation of conversion factors for fungal biomass determination in compost using ergosterol and
PLFA 18:2ω6,9. *Soil Biol. Biochem.* **36**, 57–65 (2004).

Reviewer #2:

Remarks to the Author:

Dominant taxa in soil microbial communities play outsized roles in ecosystem functioning. However, their capacities to resist and recover from climate extremes such as drought under different environmental contexts remain poorly studied. This manuscript entitled "Land management shapes drought responses of dominant soil microbial taxa across grasslands" by Lavalley et al. reports the responses of dominant bacteria and fungi immediately after drought and after a post-drought recovery period from an in situ experimental drought across 30 UK grassland sites representing a wide range of soil and climatic conditions and contrasting management intensities. The authors found that the majority of dominant bacterial and fungal taxa showed resistant or opportunistic drought strategies, while small proportions of dominant microbial taxa displayed sensitive strategies in response to drought. They also reported that intensive grassland management increased the proportion of dominant bacterial taxa with resistant and opportunistic strategies, but had the opposite impact on dominant fungal taxa. They highlighted that these links between grassland management and drought responses of dominant soil microbial taxa is a key step towards better predicting grassland ecosystem responses to drought. Overall, I think the results of this manuscript is very interesting and meaningful. However, I have several concerns about the experimental design and associated analyses:

1. It is hard to understand how the author can evaluate the resistance and resilience of dominant taxa in soil microbial communities only using data at two timepoints. Normally, long-term data from at least several timepoints before drought are needed to evaluate the microbial communities under the normal condition. Without this information, it is impossible to accurately evaluate microbial resistance and resilience to drought. Therefore, I don't think the data in this study can support the key message.
2. The authors highlight their study is across grasslands in the title. However, it seems that they don't provide any information about how many types of grasslands are studied here, though they reports that this study is carried out in 15 sites and 30 paired fields. If all 15 sites are from the same kind of grassland, the generalizability of the findings is questionable. I suggest the authors to provide more information about the grasslands in this study.
3. Line 408: the authors mention measurements of soil nematode communities and microarthropod communities, but no any results about them are reported in this manuscript. Why?
4. Line 463: Greengenes Release 13_5 is a very old version of reference database, and several new versions of Greengenes databases have been released in recent years. Thus, I suggest the authors to update their results by using the newest version of Greengenes.

All Reviewer comments are reproduced word-for-word below, with point-by-point responses in blue.

REVIEWER COMMENTS

Reviewer #1 (Remarks to the Author):

I thoroughly enjoyed reviewing this manuscript. The authors did an excellent job with the writing. It is the most well-written manuscript I have reviewed. The authors provided a generally comprehensive description of the work.

I appreciate the reasoning for the focus on dominant taxa. The takeaway that most of the dominant taxa from these soils exhibit resistant or opportunistic strategies in response to drought provides a foundation for some important theories in soil microbial ecology that have been hinted at in the literature for some time, but scarcely explicitly tested as the authors have done so here. I think this paper will help motivate other researchers in this field to make more of a point of differentiating between dominant and rare taxa and their respective roles in microbial processes and ecosystem functioning. I also appreciate the latitudinal gradient of the experimental design, which strengthens the inference power of these results and--as the authors state--make them more ecologically relevant.

I have provided minor comments in the attached word document with track changes, most of which are suggesting clarifications with phrasing or providing more detail. In particular, I would really like to see broader context added to the hypotheses and to expand on this in the Discussion. With papers like this that focus on soil microbial taxonomic composition, it's important to relay to the reader how microbial composition may influence ecosystem function. The authors have done a nice job with providing possible explanations for microbial responses to drought, land management, etc. Now they just need to bookend the story by providing some thoughts on how these microbial response strategies may affect larger scale ecosystem functions. I think the importance of this work would come through even more by making these small improvements.

We thank the reviewer for their kind words and appreciate their thoughtful review. We have copied the reviewer's comments and edits from the word document below, with point-by-point responses. This includes a similar comment to that in the final paragraph above, which we address below.

Lines 1-2: Could you make this more descriptive by adding a colon? I want to know HOW drought shaped microbial responses.

We are limited by the requirements of the journal to keep the title below 15 words and without punctuation and struggle to see how can include text to say how drought shaped microbial responses within this limit. We therefore are not able to accommodate this suggestion but do think that our current title captures the key general message of our paper.

Line 88: Even though you do so in the Methods, it would be helpful to briefly define the strategies here. I think it's also worth explicitly differentiating between resistant and resilient because many people outside of this literature often use them interchangeably.

This is a good point and have edited the text accordingly. Lines 82-89 now read:

"Using an operational approach, we identified dominant microbial taxa and classified them into three broad drought-response strategies (*i.e.*, resistant [no detectable response], opportunistic [positive response], or sensitive [negative response])¹⁴. We examined the interacting effects of climate, soil properties, and historical grassland management on dominant microbial taxa by drought-response strategy immediately following the drought and after a 60-day post-drought period²³, to capture both microbial resistance (lack of response to a perturbation) and resilience (recovery to an un-perturbed state) to drought^{24,25}."

Line 91: It would be helpful to add broader context to the hypotheses. What would each of these hypotheses, or them collectively, indicate about microbial function under climate extremes if they were supported?

Thank you for this great suggestion. We added a final paragraph to better meet the journal formatting requirements and now present brief findings and conclusions, including a sentence similar to what the reviewer has suggested above. The final paragraph of the introduction lines 98-108 now read as follows:

“Our results show that most dominant soil microbial taxa were resistant to drought, as expected. We further show that intensive grassland management increases the proportion of dominant bacterial taxa that are resistant or opportunistic in the face of drought relative to those that are sensitive, and increases the proportion of taxa that are resilient relative to those that are not resilient. However, intensive management has the opposite effect on dominant fungal taxa, increasing the proportions of sensitive and non-resilient taxa. Our finding that land management shapes the drought-response strategies of dominant soil microbial taxa has important implications for microbial community structure and function. Intensive grassland management is known to broadly favour bacteria over fungi, impacting key functions including soil carbon and nitrogen cycling^{25,26}; our results suggest this pattern may be exacerbated as droughts become more frequent and intense with climate change.”

Line 116: It would be helpful to have bacteria and fungi labels at the top or bottom of these trees so that it's immediately obvious to the reader which is which. Also, I don't recall any Methods text that describes how the trees were made.

We have added the labels to Figure 1 as suggested and added text to the Methods section describing how the plots were made (lines 491-492):

“Plots showing circular representations of the taxonomic trees were created using the GraPhlAn software tool (<https://huttenhower.sph.harvard.edu/graphlan/>).”

Line 120: delete “resistance” and change “tolerant” to “resistant”

Done (now line 131)

Line 122: delete “resilience”

Done (now line 132)

Lines 142-145: I would appreciate an Excel table of all dominant taxa, their full taxonomy, and response strategy. It's fine to not provide all this detail in the manuscript, but I think it should be made available. I'm curious (and I'm sure many other readers will be, too) to know which orders, families, etc. have resistant and sensitive taxa.

This table was provided previously as an .RData object, but we have now also included it as a .csv file along with the other data and code for this study, on figshare and at <https://github.com/soilnerd/lavallee-2023-dom-micro-drought>.

Lines 270-271: change wording to, “...opportunistic taxa to succeed under the drought treatment relative to other taxa. These opportunistic taxa...”

Done (now lines 286-287)

Lines 287-288: change wording to, “... which are thought to be comprised of mainly oligotrophs...”

Done (now lines 303-304)

Line 288: It would helpful to add in parentheses a brief description of oligotrophs as they relate to copiotrophs. Ex: generally have a slow growth strategy and thrive in nutrient-poor conditions
Thank you for the suggestion. Lines 302-305 now read:

“Further, *Verrucomicrobia* and *Acidobacteria*, which that are thought to be comprised of mainly oligotrophs (taxa that grow slowly and perform well under nutrient-poor conditions relative to copiotrophs)^{21,34,35}, had the lowest proportions of drought-sensitive taxa that were resilient.”

Line 310: These ratios are usually expressed as fungi:bacteria.
Thank you for catching this, it has been corrected. (Now line 326)

Lines 312-313: change wording to, “... as other studies have found trade-offs...”
Done (now line 329)

Line 350: change wording to, “...most of the dominant soil...”
Done (now line 365)

Lines 361-363: You’ve done a great job with the discussion, and the cherry on top would be to expand on this sentence a bit more—like I suggested in the hypotheses section of the Introduction. Just providing a few sentences about how these microbial strategy responses to drought and land management may impact particular ecosystem functions like carbon, nutrient cycling, etc. This would really help drive home the importance of this work.

Thank you very much, and we appreciate the suggestion to strengthen the discussion. We have added text to lines 373-376 as follows:

“Our results suggest the pattern of bacterial prevalence over fungi under intensive management may be reinforced or exacerbated as droughts become more frequent and intense with climate change, and potentially contribute to less efficient C and N cycling in these systems^{26,27}.”

Line 429: spell out “organic matter”
Done (now line 449)

Line 469: replace “resampled” with “rarified”
Done (now line 489)

Line 478: delete semicolon
Done (now line 500)

Lines 484-486: For clarification: were separate models run for each “dominant” OTU? I’m confused with the grouping described here. What is meant by “indices for each group”?

This is a good point, and we agree that we were not adequately clear in our description of the statistics. We have therefore edited the text to clarify this point and in lines 495-510 to read:

“All analyses were done separately for bacterial and fungal taxa in R version 4.0.2⁶⁴. We defined dominant taxa as those which were present across all 15 sites (management pairs) and represented the top 10% of taxa when ranked by relative abundance (rRNA reads). The response

of each of these dominant taxa to drought treatment was identified using a generalized linear mixed model across all experimental plot pairs with drought treatment as a fixed effect, and region/site/field as nested random effects (R package `glmmTMB`⁶⁵). For each individual model, the appropriate distribution (poisson, negative binomial, or binomial) was assumed based on diagnostics of model residuals, which were assessed using R package `DHARMA`⁶⁶. The drought-response strategy for each taxon was identified as resistant (no significant response to drought detected), sensitive (negative response), or opportunistic (positive response) using a significance level (α) of 0.05. Further statistical analysis was performed at the drought-response group level (i.e., resistant, opportunistic, sensitive, resilient, not resilient), for which group-level indices were calculated as follows: for each OTU in a given group, its relative abundance in a given sample was standardized relative to its abundance across all samples; these standardized abundances were then summed across all OTUs in a given group resulting in one value (index) per group per sample.”

Supplemental Material, Lines 86-88: I'm curious as to why pH and SWC were not also grouped in with C, N, texture, and soil temp or why each was not its own factor in the SEMs?

I don't have much experience with running SEMs myself, but are you unintentionally overweighting pH and SWC in respect to the other edaphic variables by grouping them this way?

This is a great question and appreciate the reviewer raising this point. We purposely kept soil pH and water content as individual variables in the SEM because soil water content is a reflection of the drought treatment relative to the control and is therefore important to explicitly represent. In contrast, soil pH is known to be one of the most important controls on soil microbial communities and is also managed directly in the intensive fields through liming. We also found support separating pH from the other soil properties in an early version of the SEM structure that included separate paths for each of the variables (but was not a better model in terms of explaining variance in microbial groups and was therefore not preferred relative to the final model). When modeled separately, pH consistently showed much larger standardized path coefficients than soil C, texture, and soil temperature (typically on the order of 2 – 3x the magnitude), supporting the idea that it is an important variable for explaining microbial responses and should therefore ideally be modeled separately. We have added explanation to the Supplemental Material to clarify this point, in lines 96-99 as follows:

“Two soil properties, soil water content (SWC) and pH, were not included in the “soil” variable because they were deemed important to model separately. This was done because both properties are of particular interest for this study: SWC is a reflection of the effect of the drought treatment relative to the control, and pH is known to be an important control on soil microbial communities and is also managed directly in the intensive fields.”

Reviewer #2 (Remarks to the Author):

Dominant taxa in soil microbial communities play outsized roles in ecosystem functioning. However, their capacities to resist and recover from climate extremes such as drought under different environmental contexts remain poorly studied. This manuscript entitled “Land management shapes drought responses of dominant soil microbial taxa across grasslands” by Lavalley et al. reports the responses of dominant bacteria and fungi immediately after drought and after a post-drought recovery period from an in situ experimental drought across 30 UK grassland sites representing a wide range of soil and climatic conditions and contrasting management intensities. The authors found that the majority of dominant bacterial and fungal taxa showed resistant or opportunistic drought strategies, while small proportions of dominant microbial taxa displayed sensitive strategies in response to drought. They also reported that intensive grassland management increased the proportion of dominant bacterial taxa with resistant and

opportunistic strategies, but had the opposite impact on dominant fungal taxa. They highlighted that these links between grassland management and drought responses of dominant soil microbial taxa is a key step towards better predicting grassland ecosystem responses to drought. Overall, I think the results of this manuscript is very interesting and meaningful. However, I have several concerns about the experimental design and associated analyses:

1. It is hard to understand how the author can evaluate the resistance and resilience of dominant taxa in soil microbial communities only using data at two timepoints. Normally, long-term data from at least several timepoints before drought are needed to evaluate the microbial communities under the normal condition. Without this information, it is impossible to accurately evaluate microbial resistance and resilience to drought. Therefore, I don't think the data in this study can support the key message.

We appreciate this point and agree that having long-term data would be ideal to understand variation in microbial communities through time, however, we believe the data presented in our study support the key message. Because microbial communities in soil can change dramatically through time, we designed the experiment to compare droughted plots to ambient control plots at each timepoint (and not between timepoints), as commonly done in experiments that measure resistance and resilience of microbial communities and their functioning (de Vries et al., 2012; Ingrisch and Bahn, 2018; Yi and Jackson, 2021). We purposely avoided comparisons of the droughted plots through time as we expected differences in environmental conditions and plant biomass over the season to contribute to changes over time in the droughted plots, which would have made it difficult to distinguish drought impacts from background temporal variation. The ambient control plots fluctuated through time and allowed us to capture that background temporal variation. We therefore maintain that our approach allows us to investigate resistance and resilience of microbial communities in the droughted plots relative to their paired ambient controls at the respective timepoints. However, to clarify this point and address the reviewer's point, we have edited the Methods text to clarify this aspect of the experimental design in lines 413-419:

"In total, there were 90 pairs of droughted and ambient control plots, and the three within-field replicates of each were treated appropriately in all statistical models by either including site and field as random effects or by aggregating the data at the field scale where random effects could not be modelled. At the end of the drought period, drought shelters were removed and an initial ("day 0") sampling and measurement of soil functions (in both drought and control plots) was carried out to assess the impact of the drought relative to the ambient control conditions."

Further, to support our assumption that the ambient control plots are representative of the study sites and comparable to the droughted plots in terms of key properties that are known to explain spatial variation in microbial communities (soil C and N, pH, soil temperature, soil texture), we have provided below a formal comparison of control and droughted plots with respect to those properties to show that they are similar. The PCA plot shows overlap between control (C) and drought (D) plots across sites, and the Adonis output confirms no significant differences between control and drought plots ("treatment" factor) when accounting for the hierarchical experimental design structure.

Call:
adonis(formula = sites_D0_dist ~ region * management * treatment + fieldID, data =
site_vars_control_v_drought_labels, strata = site_vars_control_v_drought_labels\$fieldID)

Blocks: strata
Permutation: free
Number of permutations: 999

Terms added sequentially (first to last)

	Df	SumsOfSqs	MeanSqs	F.Model	R2	Pr(>F)
region	2	0.22952	0.11476	-6797.8	0.27784	0.997
management	1	0.00658	0.006582	-389.9	0.00797	0.997
treatment	1	0.00377	0.003765	-223	0.00456	0.999
field	26	0.58263	0.022409	-1327.4	0.7053	0.997
region:treatment	2	0.00069	0.000343	-20.3	0.00083	0.995
management:treatment	1	0.00146	0.001458	-86.4	0.00177	0.997
region:management:treatment	2	0.00184	0.000919	-54.4	0.00222	0.996
Residuals	24	-0.00041	0.000017		0.00049	
Total	59	0.82607				1

Finally, we agree with the reviewer that we cannot make claims about the normal condition or state of the microbial communities studied here based on our experimental design, and we intended to avoid any such claims in the original version of the manuscript. However, given this concern raised by the reviewer, we went through the manuscript to identify places where this might have been mistakenly implied, and edited those for clarity as follows:

Line 231: Changed “... sensitive fungal taxa that did not return to control plot levels...” to “sensitive fungal taxa that differed from ambient control plot levels...”

Line 245: Changed “the majority were resilient (*i.e.*, fully returned to control levels within the 60-day post-drought period).” to “the majority were resilient (*i.e.*, did not differ from ambient control levels within the 60-day post-drought period).”

2. The authors highlight their study is across grasslands in the title. However, it seems that they don't provide any information about how many types of grasslands are studied here, though they reports that this study is carried out in 15 sites and 30 paired fields. If all 15 sites are from the same kind of grassland, the generalizability of the findings is questionable. I suggest the authors to provide more information about the grasslands in this study.

We appreciate the call to provide more detailed information on the variation in the grasslands studied here and have expanded upon the information previously provided in the Supplemental Material (Table S1) by adding a table with the suite of measured soil properties for ambient control plots at each site (Table S2). The values for most properties range at least two-fold across sites: from 2.9-7.6 g SOC m⁻², 0.24-0.63 g TN m⁻², 4.1-6.2 for pH (log scale), 18.3-68.5 for % sand, 0.029-0.117 for fungal:bacterial ratio, 0.34-3.8 g N m⁻² plant-available N, and 3.4-46.0 g C m⁻² microbial biomass, demonstrating high variability in key soil properties across the grassland sites used for this study. Mean soil temperature and moisture also varied widely across sites (13.6-21.3 °C and 29.3-66.3 % vol, respectively), demonstrating broad differences in climate (Table S2).

3. Line 408: the authors mention measurements of soil nematode communities and microarthropod communities, but no any results about them are reported in this manuscript. Why?

The results of the soil nematode and microarthropod measurements are complex and it would have been infeasible to explain them adequately in a single publication together with these results, which focus on the drought response strategies of dominant bacteria and fungi. They are therefore the subject of a separate manuscript currently in preparation, albeit with a very different focus to this study.

4. Line 463: Greengenes Release 13_5 is a very old version of reference database, and several new versions of Greengenes databases have been released in recent years. Thus, I suggest the authors to update their results by using the newest version of Greengenes.

We appreciate this point but do not consider this to be a major issue and would prefer not to re-run the analysis at this stage. In the figures and text, we only refer to taxa at the phyla level which will not be affected by updates in the database. Therefore, using a different taxonomic database to annotate our sequences will not affect our interpretation of results. Additionally, we note that the suggested last version of the Greengenes database is also now several years old, so ideally we would use a different database (Silva). That said, we are happy to re-run the analyses at the reviewer's insistence.

References

- de Vries, F.T., Liiri, M.E., Bjørnlund, L., Bowker, M.A., Christensen, S.S., Setälä, H.M., Bardgett, R.D., Bardgett, 2012. Land use alters the resistance and resilience of soil food webs to drought. *Nature Climate Change* 2, 276–280. doi:10.1038/nclimate1368
- Ingrisch, J., Bahn, M., 2018. Towards a Comparable Quantification of Resilience. *Trends in Ecology and Evolution* 33, 251–259. doi:10.1016/j.tree.2018.01.013
- Yi, C., Jackson, N., 2021. A review of measuring ecosystem resilience to disturbance. *Environmental Research Letters* 16. doi:10.1088/1748-9326/abdf09

Reviewers' Comments:

Reviewer #1:

Remarks to the Author:

The authors have adequately responded to the reviewer comments. From my reviewer perspective the manuscript is acceptable for publication.

Reviewer #2:

Remarks to the Author:

After reading through the revised manuscript and response to reviewers, I think the authors make a big effort to revise the manuscript. The additions in background information of grassland and methods of resistance and resilience bring a great improvement. Although the authors can't fully address the concerns about the resistance and resilience of communities on long time scales, their study provide insight in the resistance and resilience of soil dominant taxa in response to drought. Another thing is that it would certainly increase the reliability of the results in this study if the authors re-run the analyses by using new version of reference database (i.e., Silva or Geengenes2). I have nothing to add.

Point-by-point response to reviewer comments for “Land management shapes drought responses of dominant soil microbial taxa across grasslands” by Lavallee et al.

Reviewer #1 (Remarks to the Author):

The authors have adequately responded to the reviewer comments. From my reviewer perspective the manuscript is acceptable for publication.

Response: We thank the reviewer for their thoughtful consideration.

Reviewer #2 (Remarks to the Author):

After reading through the revised manuscript and response to reviewers, I think the authors make a big effort to revise the manuscript. The additions in background information of grassland and methods of resistance and resilience bring a great improvement. Although the authors can't fully address the concerns about the resistance and resilience of communities on long time scales, their study provide insight in the resistance and resilience of soil dominant taxa in response to drought. Another thing is that it would certainly increase the reliability of the results in this study if the authors re-run the analyses by using new version of reference database (i.e., Silva or Geengenes2). I have nothing to add.

Response: We thank the reviewer for their thoughtful comments and appreciate their responses to our revisions. After careful consideration we feel that re-analysis using an updated database would not provide any significant additional insight (only some of the namings would change), especially in light of the effort and time required to edit the results and figures, therefore we prefer to leave as is.